# Caspase-1 initiates apoptosis in the absence of gasdermin D

Kohsuke Tsuchiya[1,2], Shinsuke Nakajima[1], Shoko Hosojima[1], Dinh Thi Nguyen [3], Tsuyoshi Hattori [3], Thuong Manh Le[3], Osamu Hori[3], Mamunur Rashid Mahib[1], Yoshifumi Yamaguchi [4], Masayuki Miura [5], Takeshi Kinoshita[1], Hiroko Kushiyama[1], Mayumi Sakurai[1], Toshihiko Shiroishi[6] & Takashi Suda[1]

Caspase-1 activated in inflammasomes triggers a programmed necrosis called pyroptosis, which is mediated by gasdermin D (GSDMD). However, GSDMD-deficient cells are still susceptible to caspase-1-mediated cell death. Therefore, here, we investigate the mechanism of caspase-1-initiated cell death in GSDMD-deficient cells. Inflammasome stimuli induce apoptosis accompanied by caspase-3 activation in GSDMD-deficient macrophages, which largely relies on caspase-1. Chemical dimerization of caspase-1 induces pyroptosis in GSDMD-sufficient cells, but apoptosis in GSDMD-deficient cells. Caspase-1-induced apoptosis involves the Bid-caspase-9-caspase-3 axis, which can be followed by GSDME-dependent secondary necrosis/pyroptosis. However, Bid ablation does not completely abolish the cell death, suggesting the existence of an additional mechanism. Furthermore, cortical neurons and mast cells exhibit little or low GSDMD expression and undergo apoptosis after oxygen glucose deprivation and nigericin stimulation, respectively, in a caspase-1- and Bid-dependent manner. This study clarifies the molecular mechanism and biological roles of caspase-1-induced apoptosis in GSDMD-low/null cell types.

[1] Division of Immunology and Molecular Biology, Cancer Research Institute, Kanazawa University, Kakuma-machi, Kanazawa 920-1192, Japan. [2] Institute for Frontier Science Initiative (InFiniti), Kanazawa University, Kakuma-machi, Kanazawa 920-1192, Japan. [3] Department of Neuroanatomy, Graduate School of Medical Sciences, Kanazawa University, 13-1 Takara-Machi, Kanazawa 920-8640, Japan. [4] Institute of Low Temperature Science, Hokkaido University, Kita-19, Nishi-8, Kita-ku, Sapporo 060-0819, Japan. [5] Department of Genetics, Graduate School of Pharmaceutical Sciences, The University of Tokyo, 7-3-1 Hongo, Bunkyo-ku, Tokyo 113-0033, Japan. [6] Mammalian Genetics Laboratory, Genetic Strains Research Center, National Institute of Genetics, 1111 Yata, Mishima, Shizuoka 411-8540, Japan. Correspondence and requests for materials should be addressed to K.T. (email: ktsuchiya@staff.kanazawa-u.ac.jp) or to T.S. (email: sudat@staff.kanazawa-u.ac.jp)

Caspases are a family of cysteine proteases that mediate regulated cell death, including apoptosis and pyroptosis[1]. To date, 13 caspases have been identified in mammals (these exclude caspase-11, which is now called mouse caspase-4, and caspase-13, which was a misnomer for bovine caspase-4). While most caspases act in apoptosis as initiators or effectors, several inflammatory caspases, such as caspase-1, caspase-4 (caspase-11 in mice), and caspase-5, trigger pyroptosis[2–4]. Apoptosis and pyroptosis are different morphologically, functionally, and mechanistically. For example, while the loss of membrane integrity is a late event in apoptosis, pyroptosis is a lytic cell death accompanied by rapid cell-membrane rupture[2–5]. Caspase-3, -7, and -6 serve as effector caspases in apoptosis and induce phosphatidylserine (PS) externalization, cell shrinkage, membrane blebbing, and DNA fragmentation by cleaving so-called death substrates[2,3]. In pyroptosis, pores are formed in the cell membrane, which cause water influx and cell swelling, resulting in cell-membrane rupture[4,6]. Pyroptotic cells release cellular contents, including inflammatory cytokines and damage-associated molecular patterns (DAMPs), so pyroptosis is thought to be more inflammatory and immunogenic than apoptosis[2–4]. Pyroptosis has been suggested to contribute to host defense against microbial infections[7–9], while it has also been implicated in the pathogenesis of diseases, such as endotoxin shock and acquired immune deficiency syndrome[10,11].

Caspase-1 is activated in inflammasomes, which are multi-protein oligomers composed of a certain intracellular pattern recognition receptor (PRR), the adaptor apoptosis-associated speck-like protein containing a caspase recruitment domain (ASC), and pro-caspase-1[4,12]. Inflammasome-forming PRRs, including NLR family pyrin domain containing 3 (NLRP3), NLR family CARD domain containing 4 (NLRC4) and absent in melanoma 2 (AIM2), respond to specific ligands or activators that facilitate the assembly of inflammasomes, in which pro-caspase-1 molecules are brought into close proximity, resulting in caspase-1 activation[4,12,13]. While NLRP3 and AIM2's binding to pro-caspase-1 requires ASC, NLRC4 can bind to pro-caspase-1 directly, thereby inducing pyroptosis independent of ASC[4,12,14]. On the other hand, caspase-11, -4, and -5 directly bind to cytoplasmic lipopolysaccharide (LPS), leading to the oligomerization and activation of these caspases; thus, these caspases serve as sensor molecules for intracellular LPS[15–17].

Recently, gasdermin D (GSDMD) was identified as a critical mediator of pyroptosis[18–20]. Active caspase-1, -4, -5, and -11 cleave GSDMD within a linker between its N-terminal and C-terminal domains. After cleavage, the N-terminal domain forms pores in the cell membrane to cause pyroptosis[21–23]. The C-terminal domain of GSDMD inhibits the pore-forming activity of the N-terminal domain[21–23], and it is speculated that GSDMD's cleavage by inflammatory caspases relieves the N-terminal domain from this autoinhibition. The cell death caused by intracellular LPS was shown to be completely abrogated in the absence of GSDMD[18–20], suggesting that GSDMD is the sole mechanism by which the LPS-sensing caspases (caspase-11, -4, and -5) trigger cell death. By contrast, caspase-1 has been suggested to participate in both GSDMD-dependent and GSDMD-independent cell death pathways, given that inflammasome stimuli can induce cell death in GSDMD-deficient cells[19,20]. However, the GSDMD-independent cell death pathway is not well understood. Moreover, it is still controversial whether this pathway relies on caspase-1, because inflammasome formation results in not only the activation of caspase-1 but also that of caspase-8 via ASC speck formation[24]. In fact, ASC-dependent caspase-8-mediated apoptosis was observed when caspase-1-deficient cells were stimulated with inflammasome activators[25]. Hence, it is also conceivable that caspase-8 mediates the GSDMD-independent cell death observed after inflammasome formation in the absence of GSDMD. Thus, based on the current evidence, it remains unclear whether and how caspase-1 induces cell death independent of GSDMD.

In this study, we investigate caspase-1-induced cell death pathways. Our results clearly demonstrate that caspase-1 can initiate apoptosis in the absence of GSDMD. Moreover, we find that the caspase-1-induced apoptosis involves the Bid-caspase-9-caspase-3 axis, which can elicit secondary necrosis/pyroptosis mediated by GSDME (also known as DFNA5). Furthermore, GSDMD appears to inhibit the caspase-1-initiated apoptosis in a manner involving its pore-forming activity. Thus, this study reveals the complex caspase-1-induced cell death mechanisms. Notably, in early studies, caspase-1 was considered a pro-apoptotic caspase;[26–29] however, caspase-1's induction of apoptosis is not as well established as its role in pyroptosis. This study also sheds light on this neglected aspect of caspase-1.

## Results

**Caspase-1 mediates apoptosis in GSDMD-deficient macrophages.** To determine whether caspase-1 induces cell death in the absence of GSDMD, thioglycolate-elicited peritoneal macrophages (TEPMs) and bone marrow-derived macrophages (BMMs) from wild-type (WT), $Gsdmd^{-/-}$, $Pycard$ (the gene for ASC)$^{-/-}$, and $Casp1^{-/-}$ mice were stimulated with inflammasome activators, and cell death was assessed by lactate dehydrogenase (LDH) release from the cells. Transfection with poly (dA:dT), which activates the AIM2 inflammsome[13], induced rapid LDH release from WT TEPMs indicating that pyroptosis was induced, but not from $Pycard^{-/-}$ or $Casp1^{-/-}$ TEPMs within the same time window (Fig. 1a). Importantly, a delayed but significant LDH release was observed when the $Gsdmd^{-/-}$ TEPMs were transfected with poly(dA:dT). Similarly, when BMMs were infected with *Salmonella* (*S.*) Typhimurium, which activates the NLRC4 inflammasome[30], the $Gsdmd^{-/-}$ BMMs released LDH more rapidly than did the $Casp1^{-/-}$ BMMs (Fig. 1b). These results indicated that inflammasome formation can lead to caspase-1 dependent cell death even in the absence of GSDMD. As expected, two hours after *S.* Typhimurium infection, the WT BMMs showed a pyroptotic morphology (Fig. 1c). On the other hand, apoptotic morphological changes, such as membrane blebbing[2,3], were induced in the $Gsdmd^{-/-}$ BMMs, but not in the WT or $Casp1^{-/-}$ BMMs (Fig. 1c). The p10 subunit of mature caspase-1 was detected in culture supernatants plus cell lysates of the WT and $Gsdmd^{-/-}$ BMMs after *S.* Typhimurium infection (Fig. 1d). Notably, *S.* Typhimurium infection strongly induced the activation of caspase-3, caspase-8, and caspase-9 in the $Gsdmd^{-/-}$ BMMs (Fig. 1d). By contrast, cleaved forms of these caspases were not or only barely detected in the WT and $Casp1^{-/-}$ BMMs. These results suggested that GSDMD-deficient macrophages undergo caspase-1-dependent apoptosis after inflammasome formation.

To exclude the possible involvement of the ASC-caspase-8 pathway in the *S.* Typhimurium-induced cell death of GSDMD-deficient macrophages, we used RAW264.7 cells, a macrophage-like cell line that lacks ASC expression (Supplementary Fig. 1)[31]. Just like the BMMs, while WT RAW264.7 cells died by pyroptosis after *S.* Typhimurium infection, the $Gsdmd$ knockout (KO) RAW264.7 cell clones exhibited apoptotic features including membrane blebbing and caspase-3 activation (Fig. 1e–g). These responses were not seen in $Casp1$/$Gsdmd$-double KO (DKO) cells. In addition, consistent with a recent report[32], caspase-3 activation, apoptotic morphological changes, and LDH release were observed in the $Gsdmd$-KO cells after treatment with Val-boroPro, a recently described inducer of caspase-1 activation[33]

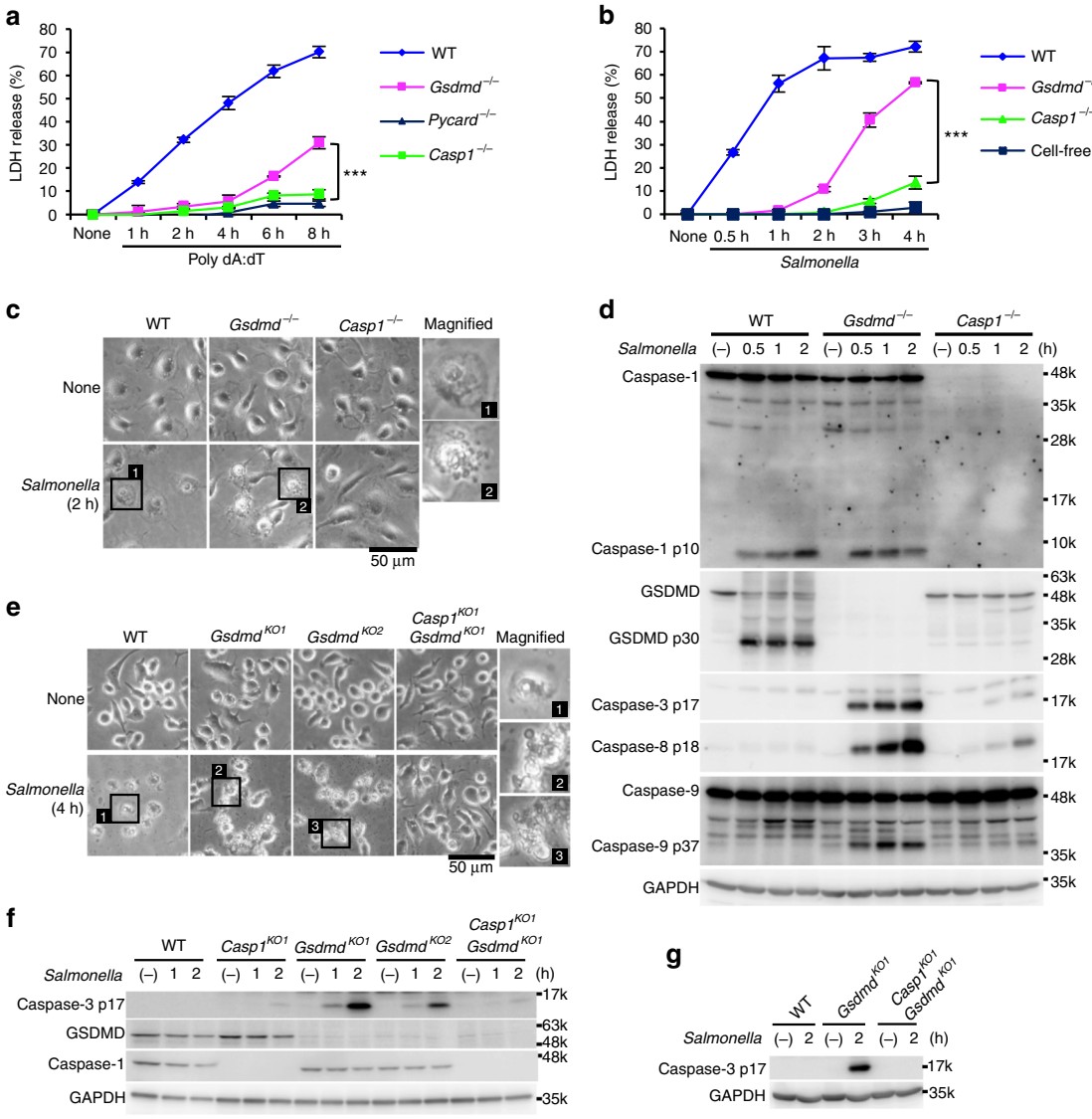

**Fig. 1** GSDMD-deficient macrophages undergo apoptosis after inflammasome activation. **a–d** TEPMs (**a**) and BMMs (**b–d**) of the indicated genotypes were transfected with 150 ng of Poly dA:dT (**a**) or infected with *S*. Typhimurium at an MOI of 5 (**b–d**). Bacteria were washed out (**c**) or not washed out (**b, d**), and gentamicin was added to the BMM cultures at 1 h after infection. Cell death was monitored by LDH release assay (**a, b**). Graphs depict the mean ± SD of triplicate cultures. ***$p < 0.001$ (Bonferroni's multiple comparisons test). Microscopic images (**c**). Western blot detection of activated caspases in culture supernatants plus cell lysates (Sup + CL) (**d**). **e–g** RAW264.7 cells of the indicated genotypes were infected with *S*. Typhimurium at an MOI of 40. Bacteria were washed out, and gentamicin was added to the cultures at 2 h after infection (**e**). *S*. Typhimurium-infected *Gsdmd*-KO RAW264.7 cells showed apoptotic morphological changes. Western blot detection of caspase-3 p17 in cell lysates (**f**) and Sup + CL (**g**). Data are from one representative of three biologically independent experiments with similar results (**a–g**). Source data are provided as a Source Data file. (See also Supplementary Figs. 1 and 2.)

(Supplementary Fig. 2a–c). Importantly, these responses were also abrogated in *Casp1/Gsdmd*-DKO cells (Supplementary Fig. 2a–c). These results collectively indicated that caspase-1 can trigger apoptosis in the absence of GSDMD.

**Caspase-1 initiates apoptosis in GSDMD-deficient cells**. To examine caspase-1's ability to induce apoptosis directly, we constructed a recombinant mouse procaspase-1, and procaspase-8 as a control, fused with the three tandem N-terminal Fv domains (hereafter Fv3-mCasp1 and Fv3-mCasp8, respectively), which interact with the bivalent chemical compound AP20187 (Fig. 2a)[34,35]. Fv3-mCasp1 and Fv3-mCasp8 are oligomerized and activated in the presence of AP20187. Colon-26 murine colorectal carcinoma cells, which express GSDMD but not ASC (Supplementary Fig. 1), were stably transfected with Fv3-mCasp1 (CL26-

iCasp1 cells) or Fv3-mCasp8 (CL26-iCasp8 cells). As expected, the former and the latter underwent necrotic and apoptotic cell death, respectively, after AP20187 treatment (Supplementary Fig. 3a, Supplementary Movie 1). Depletion (knockdown and KO) of GSDMD significantly delayed but did not completely block the LDH release from AP20187-treated CL26-iCasp1 cells (Fig. 2b–d, Supplementary Fig. 3b). AP20187 induced caspase-3 activation in the *Gsdmd*-KO but not WT cells (Fig. 2c and Supplementary Fig. 3d). In addition, PS externalization and plasma membrane blebbing before plasma membrane permeabilization, hallmarks of apoptosis[2,3], was observed in the AP20187-treated *Gsdmd*-KO CL26-iCasp1 cells (Fig. 2e, f, and Supplementary Fig. 3c, Supplementary Movie 2).

Moreover, the reconstitution of GSDMD expression in *Gsdmd*-KO CL26-iCasp1 cells abolished the activation of caspase-3

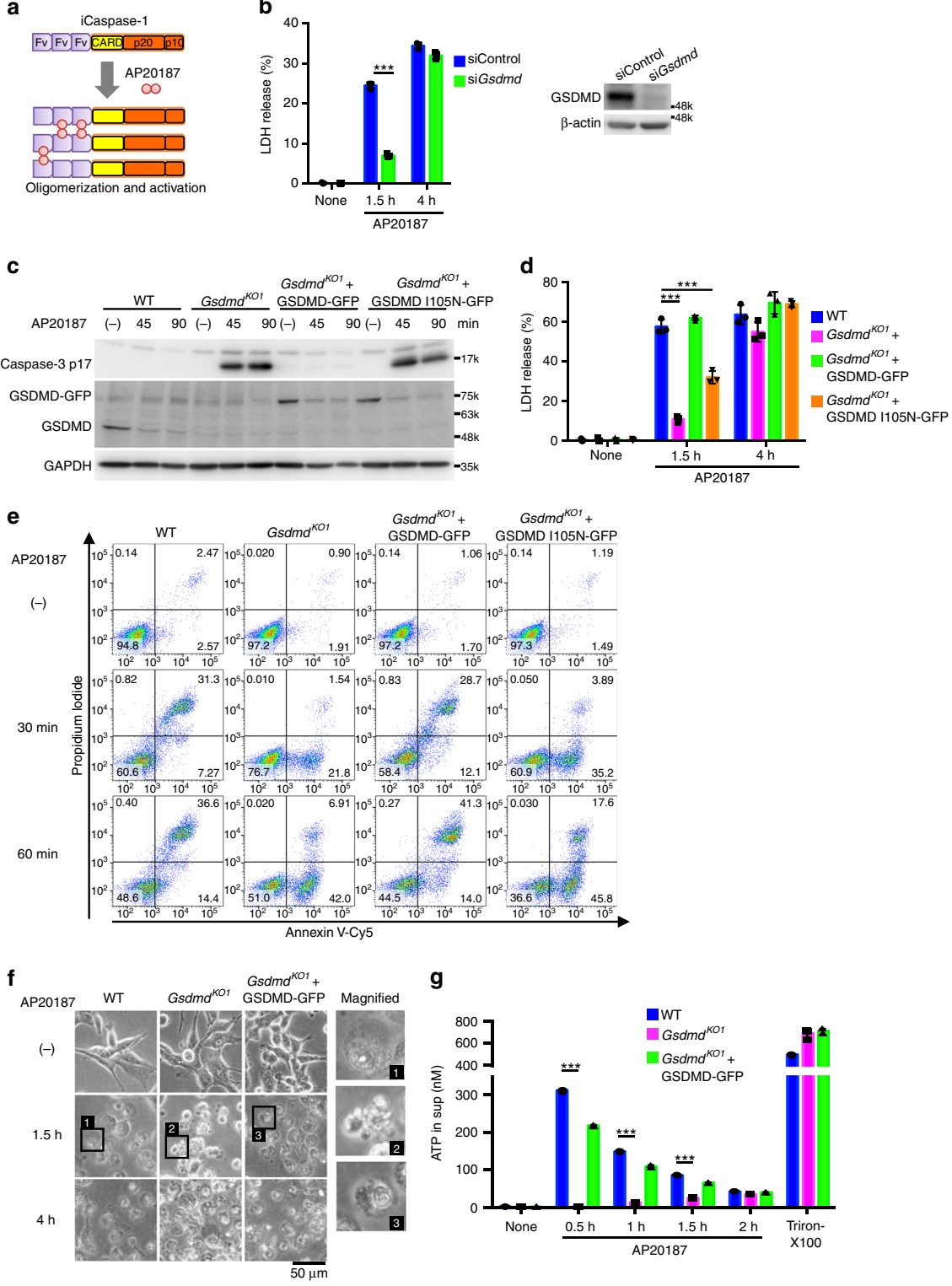

**Fig. 2** Caspase-1 initiates apoptosis in the absence of functional GSDMD. **a** Scheme of the Fv3-mCasp1 protein. AP20187 binds to the Fv domain, leading to the oligomerization and activation of Fv3-mCasp1. **b** CL26-iCasp1 cells were transfected with control siRNA or *Gsdmd* siRNA. Two days after transfection, the cells were treated with 50 nM AP20187 for the indicated times, and cell death was monitored by LDH release assay. GSDMD was detected by Western blotting. **c–g** CL26-iCasp1 cells of the indicated genotypes transduced or not transduced with GSDMD-GFP or GSDMD I105N-GFP were treated with 50 nM AP20187. Cleaved caspase-3 was detected by Western blotting (**c**). LDH release (**d**). PI uptake and PS exposure analyzed by flow cytometry (e, *n* = 2). Microscopic images (**f**). ATP in culture supernatants was measured (**g**). In **b**, **d**, and **g**, graphs depict the mean ± SD of triplicate cultures, and individual data values are plotted. Statistical significance was determined using an unpaired Student's *t*-test (**b**) or Bonferroni's multiple comparisons test (**d**, **g**). ***p < 0.001. Data are from one representative of two (**e**), three (**c**, **f**, and **g**), or four (**b**, **d**) independent experiments with similar results. Source data are provided as a Source Data file. (See also Supplementary Fig. 3.)

(Fig. 2c and Supplementary Fig. 3e) and restored the early LDH release (1.5 h after AP20187 treatment) (Fig. 2d). GSDMD-reconstituted cells, like WT cells, exhibited a necrotic morphology with a loss of cell membrane integrity quickly after AP20187 treatment (Fig. 2e, f). These results clearly indicated that caspase-1 can initiate both pyroptosis and apoptosis, depending on the expression of GSDMD. Interestingly, reconstitution with the I125N mutant of GSDMD, whose pore-forming activity is severely attenuated[19,23], did not have these effects (Fig. 2c, e), suggesting that the pore-forming activity of GSDMD determines whether pyroptosis or apoptosis will occur downstream of caspase-1 activation.

We also found that ATP, a DAMP that induces chemotaxis and activation of the NLRP3 inflammasome in macrophages[2,4], was released from GSDMD-expressing cells, but not from Gsdmd-KO cells, immediately after AP20187 treatment (Fig. 2g). The ATP release occurred prior to LDH release (Supplementary Fig. 3f), suggesting that ATP is released through the GSDMD-formed pores before cell lysis.

**Caspase-1-induced apoptosis depends on caspase-3.** Although caspase-3 plays a central role in apoptosis, caspase-3-independent apoptosis has also been described in previous studies[36,37]. Using SCAT3, a fluorescence resonance energy transfer (FRET) probe that detects caspase-3 activation[38], we observed that caspase-3 was activated along with the apoptotic morphological changes induced by AP20187 in Gsdmd-KO CL26-iCasp1 cells (Supplementary Fig. 4a). Moreover, the ablation of caspase-3 almost completely abrogated the LDH release and PS exposure induced by AP20187 (Fig. 3a–c and Supplementary Fig. 4b), indicating that the caspase-1-induced apoptosis depends on caspase-3. Consistent with the results obtained using Gsdmd-KO CL26-iCasp1 cells, the apoptotic morphological changes induced in Gsdmd-KO RAW264.7 cells by S. Typhimurium or Val-boroPro were also abolished when caspase-3 was depleted (Fig. 3d, e and Supplementary Fig. 4c), indicating that caspase-1 induces caspase-3-dependent apoptosis in macrophages in the absence of GSDMD.

**GSDME mediates necrosis after caspase-1-induced apoptosis.** Given that caspase-1 activation induces apoptosis in the absence of GSDMD, LDH release from Gsdmd-KO cells induced by caspase-1 activation is likely to be a result of secondary necrosis. Because LDH release was observed as rapid as 2 h after S. Typhimurium infection in Gsdmd[−/−] BMMs (Fig. 1b) and 1.5 h after AP20187 treatment of Gsdmd-KO CL26-iCasp1 cells (Fig. 2b), one may wonder if caspase-1-initiated apoptosis might proceed to the secondary necrosis stage more rapidly than regular (e.g. caspase-8-initiated) apoptosis. However, the time intervals between caspase-3/7 activation and LDH release in these instances of caspase-1-initiated apoptosis was similar to the intervals in cspase-8-initiated apoptosis induced by Fas ligand plus cycloheximide treatment in Gsdmd[−/−] BMMs and that induced by AP20187 treatment in CL26-iCasp8 cells (Supplementary Fig. 4d–g). Thus, the speed at which the cell death process proceeds to secondary necrosis was similar between caspase-1 and caspase-8-initiated apoptosis.

Recently, caspase-3 was demonstrated to cleave GSDME, to induce a secondary necrosis/pyroptosis[39,40]. Indeed, GSDME was cleaved in a caspase-3-dependent manner in Gsdmd-KO CL26-iCasp1 cells after AP20187 treatment (Fig. 3a). We found that the KO of GSDME in Gsdmd-KO CL26-iCasp1 cells severely reduced the cell membrane damage observed within 4 h after AP20187 treatment, as determined by LDH release and staining with the cell-impermeant fluorescent dyes propidium iodide and YO-

PRO-1 (Fig. 3f, g; Supplementary Fig. 5a–d; Supplementary Movie 3), while GSDME ablation did not affect the PS exposure, membrane blebbing, or caspase-3 activation in the Gsdmd-CL26-iCasp1 cells (Fig. 3g and Supplementary Fig. 5b–d). However, Gsdmd/Gsdme-DKO CL26-iCasp1 cells became necrotic, and extensive LDH release was observed by 24 h after AP20187 treatment (Supplementary Fig. 5e–g). These results indicated that GSDME mediates a rapid secondary necrosis/pyroptosis after the caspase-1-induced apoptosis, but is not essential for the cell lysis at later times. GSDME was also cleaved in GSDMD-deficient BMMs after S. Typhimurium infection (Supplementary Fig. 5h). However, the role of GSDME in BMM apoptosis has been controversial[39,41]. RAW264.7 cells expressed little GSDME (Supplementary Fig. 1), and thus this cell line is not suitable to test the role of GSDME in the secondary necrosis/pyroptosis. Thus, further experiments using BMMs from Gsdmd/Gsdme-DKO mice are required to determine the contribution of GSDME in the secondary necrosis/pyroptosis of GSDMD-deficient macrophages.

**Caspase-1-induced apoptosis involves caspase-9.** Next, we examined the possible involvement of other caspases in the caspase-1-initiated apoptosis in GSCMD-deficient cells. The proteolytic activation of caspase-2, -7, -8, and -9 was observed in the AP20187-treated Gsdmd-KO CL26-iCasp1 cells (Fig. 4a, b). The cleavage of caspase-8 (generation of the p18 fragment) and caspase-2 (generation of the p30 fragment) observed in Gsdmd-KO CL26-iCasp1 after AP20187 treatment and in RAW264.7 cells after S. Typhimurium infection was almost completely abrogated in the Gsdmd/Casp3-DKO cell lines (Fig. 4c–e). The cleavage of caspase-7 (generation of the p20 fragment) and caspase-9 (generation of the p37 fragment) was also reduced in the Gsdmd/Casp3-DKO cells compared to Gsdmd-KO cells; however, residual activation of caspase-7 and -9 was reproducibly observed in the Gsdmd/Casp3-DKO cells (Fig. 4c, e). These results indicated that caspase-2 and -8 are activated downstream of caspase-3, whereas caspase-7 and -9 can be activated both independently of (upstream of or in parallel with) and downstream of caspase-3 in Gsdmd-KO cells under these experimental conditions.

We next set out to identify caspases involved in the caspase-1-induced apoptosis. Depleting caspase-2, -6, or -7 from the Gsdmd-KO CL26-iCasp1 cells did not affect the apoptosis induced by AP20187 (Fig. 4f, g and Supplementary Fig. 6a). Gsdmd/Rip3k-DKO CL26-iCasp1 cells were then used to generate Casp8-KO cells to prevent necroptosis induction. The Gsdmd/Rip3k-DKO and Gsdmd/Rip3k/Casp8-triple KO cell lines underwent apoptosis normally after AP20187 treatment (Fig. 4h, i and Supplementary Fig. 6b), indicating that the caspase-1-induced apoptosis can proceed independently of these caspases. By contrast, depleting caspase-9 from the Gsdmd-KO CL26-iCasp1 cells markedly reduced the activation of caspase-3 and GSDME, PS exposure, and LDH release induced by AP20187 treatment (Fig. 5a–d), which were restored by complementation with the Casp9 cDNA (Fig. 5e and Supplementary Fig. 7a, b), indicating that caspase-9 mediates the apoptosis initiated by caspase-1.

**Bid plays a critical role in caspase-1-induced apoptosis.** A central mechanism in the activation of caspase-9 is the formation of an apoptosome after the release of cytochrome c from mitochondria into the cytoplasm[2,3]. Cytochrome c release into the cytosol, mitochondrial depolarization, and formation of an Apaf-1-caspase-9 complex were observed in Gsdmd-KO CL26-iCasp1 cells after AP20187 treatment (Fig. 6a and Supplementary Fig. 8a,

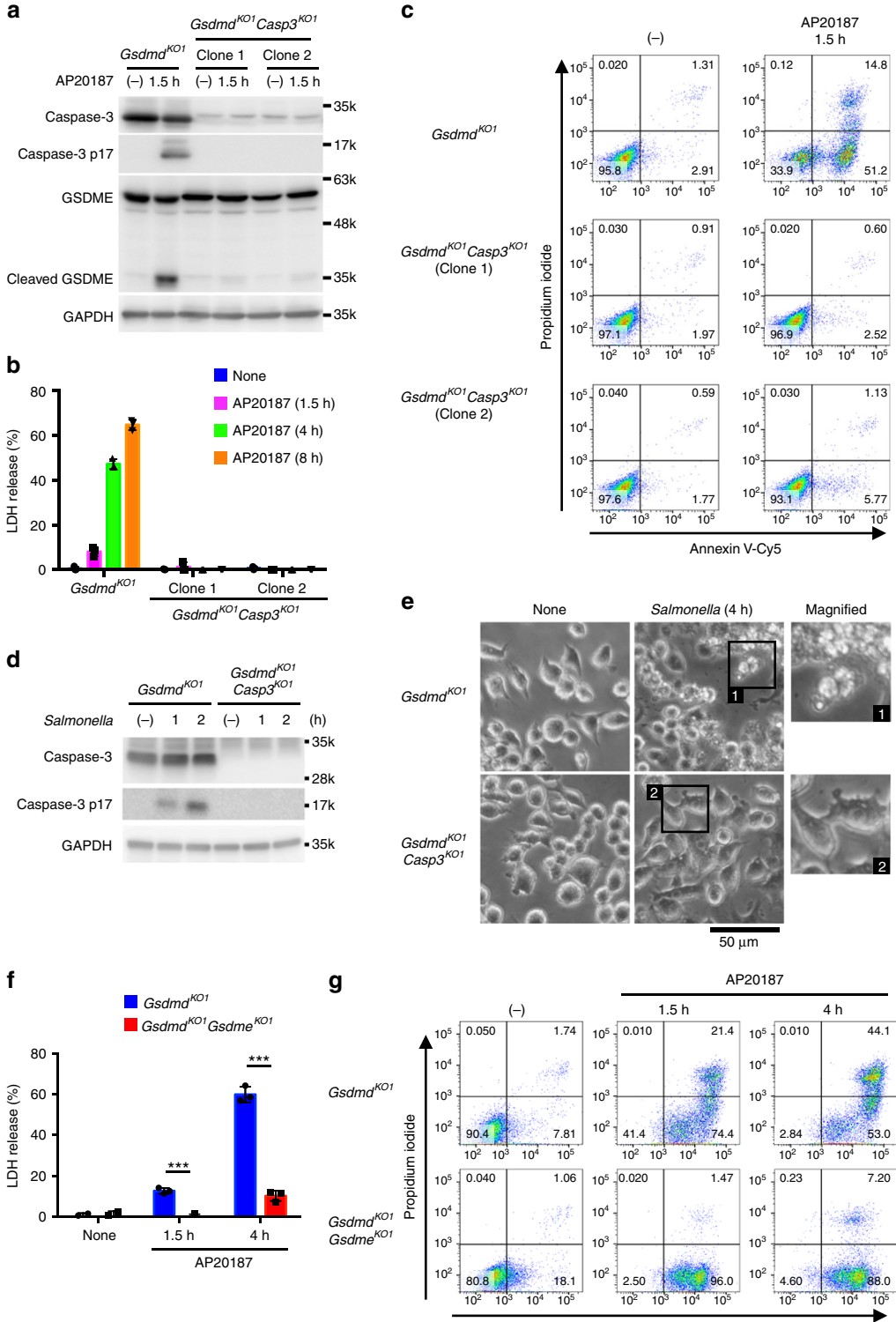

**Fig. 3** Caspase-1-induced apoptosis depends on caspase-3. **a–c** *Gsdmd*-KO and *Gsdmd/Casp3*-DKO CL26-iCasp1 cells were treated with AP20187 for the indicated times. Western blot detection of caspase-3 and GSDME in cell lysates (**a**). LDH release from the AP20187-treated cells (**b**). PI uptake and PS exposure analyzed by flow cytometry (**c**). Percentages of Annexin V+ cells are shown in Supplementary Fig. 4b. **d, e** *Gsdmd*-KO and *Gsdmd/Casp3*-DKO RAW264.7 cells were infected with *S.* Typhimurium as in Fig. 1f (**d**) or Fig. 1e (**e**). Caspase-3 in cell lysates detected by Western blotting (**d**). Microscopic images (**e**). **f, g** *Gsdmd*-KO and *Gsdmd/Gsdme*-DKO CL26-iCasp1 cells were treated with AP20187, and cell death was monitored by LDH release (**f**) and flow cytometry (**g**). Percentages of Annexin V+ PI+ cells and Annexin V+ PI− cells are shown in Supplementary Fig. 5d. In **b** and **f**, graphs depict the mean ± SD of triplicate cultures, and individual data values are plotted (**b**, **f**). Statistical significance was determined using an unpaired Student's *t*-test (**f**). ***p < 0.001. Data are from one representative of three independent experiments with similar results (**a–g**). Source data are provided as a Source Data file. (See also Supplementary Figs. 4 and 5.)

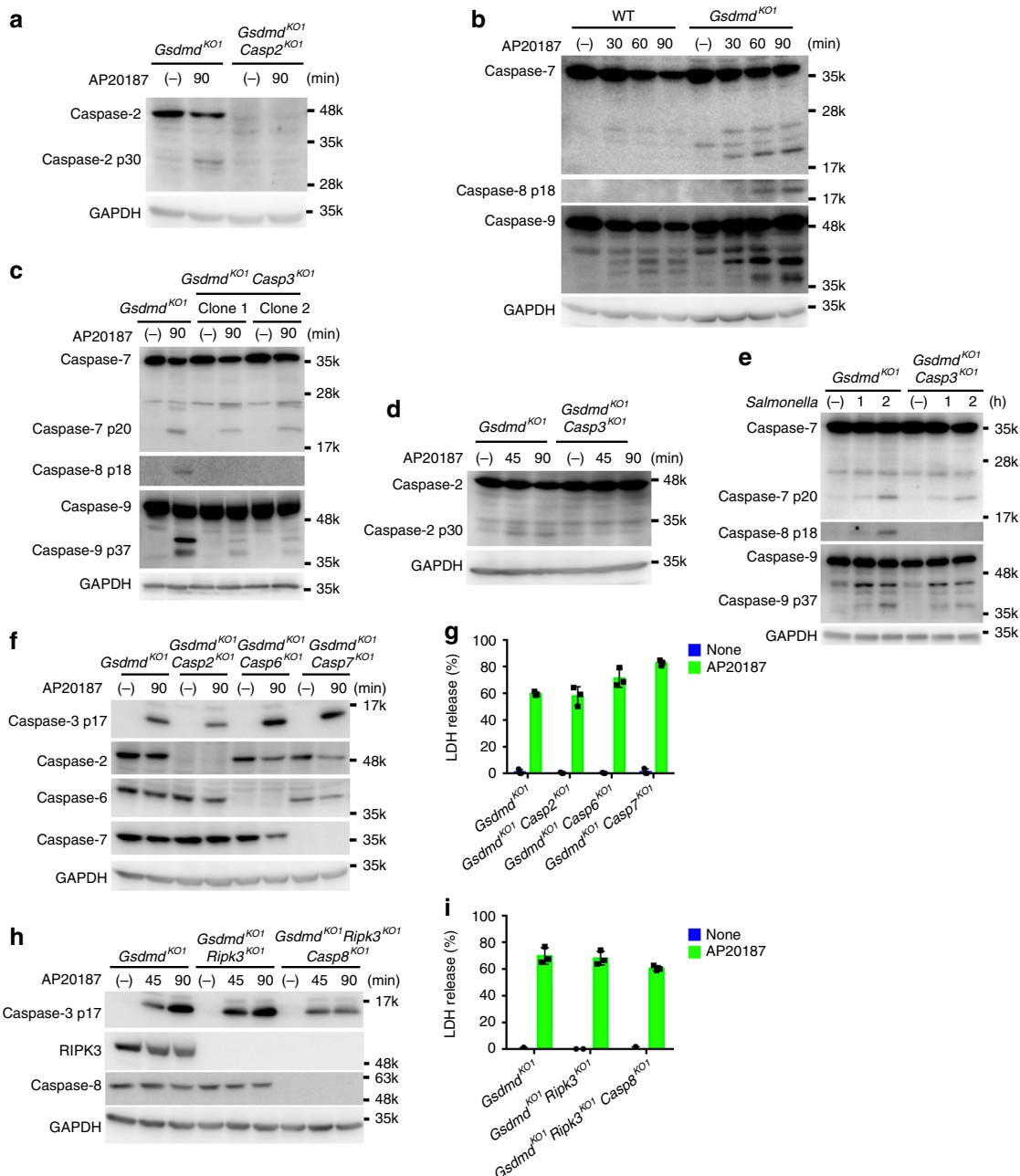

**Fig. 4** Caspase-2, 6, 7, and 8 are dispensable for caspase-1-induced apoptosis. **a** *Gsdmd*-KO and *Gsdmd/Casp2*-DKO CL26-iCasp1 cells were stimulated with 50 nM AP20187 for 1.5 h, and cleaved caspase-2 was detected by Western blotting. **b** Western blot detection of cleaved caspases in WT and *Gsdmd*-KO CL26-iCasp1 cells treated with AP20187 for the indicated times. **c, d, e** *Gsdmd*-KO and *Gsdmd/Casp3*-DKO CL26-iCasp1 cells (**c, d**) and RAW264.7 cells (**e**) were treated with AP20187 and infected with *S*. Typhimurium as in Fig. 1f, respectively. Cleaved caspases were detected by western blotting. **f–i** CL26-iCasp1 cells of the indicated genotypes were treated with AP20187 for the indicated times (**f, h**) or 4 h (**g, i**). Western blot detection of cleaved caspase-3 and procaspases in the cells (**f, h**). LDH release from the cells (**g, i**). Graphs depict the mean ± SD of triplicate cultures, and individual data values are plotted. Data are from one representative of three independent experiments with similar results (**a–i**). Source data are provided as a Source Data file. (See also Supplementary Fig. 6)

b). Bcl-2 homology 3 (BH3)-only proteins are known to cause cytochrome c release[2]. Among the BH3-only family members, Bid was previously shown to be cleaved into its active form, truncated Bid (tBid) by caspase-1 in vitro[42]. We observed the processing of Bid into tBid in WT and *Gsdmd*-KO CL26-iCasp1 cells treated with AP20187, and tBid was released from the cells in a GSDMD-dependent manner (Fig. 6b). Remarkably, cytochrome c release, mitochondrial depolarization, and Apaf-1-caspase-9

complex formation were not induced by AP20187 until 1 h after AP20187 treatment if Bid was ablated (Fig. 6a, Supplementary Fig. 8a, b, and Supplementary Fig. 9a). Moreover, the depletion of Bid severely impaired the LDH release from *Gsdmd*-KO CL26-iCasp1 cells at least up to 4 h after AP20187 treatment (Fig. 6c). Under the same conditions, caspase-3 activation and PS exposure were also severely diminished until 1.5 h after AP20187 treatment (Fig. 6d and Supplementary Fig. 9b, c). Furthermore, knockdown

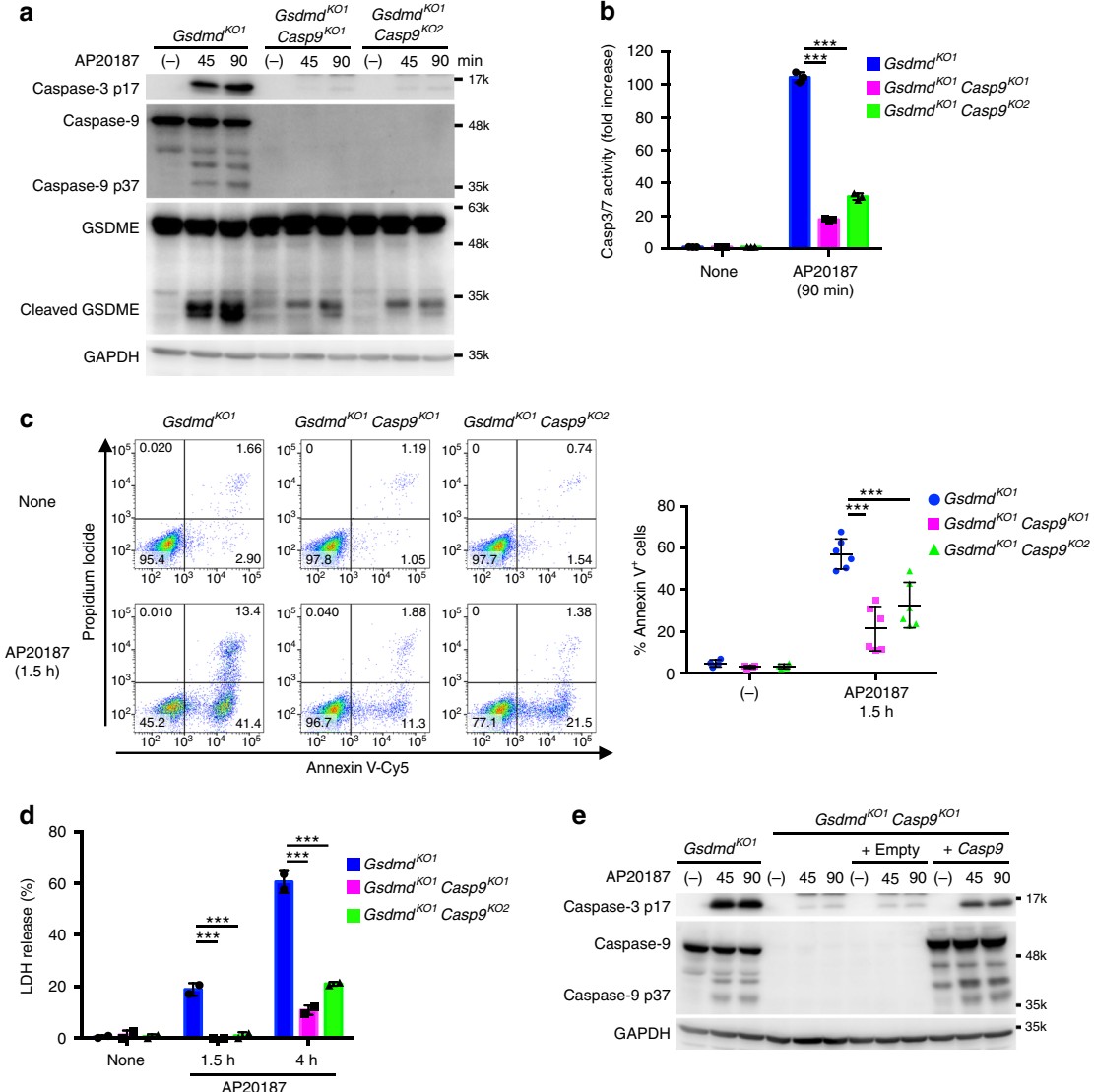

**Fig. 5** Caspase-9 is involved in caspase-1-induced apoptosis. **a–d** *Gsdmd*-KO and *Gsdmd/Casp9*-DKO CL26-iCasp1 cells were treated with 50 nM AP20187 for the indicated times. Western blot detection of cleaved caspases and GSDME in cell lysates (**a**). Caspase-3/7 activity in cell lysates determined using a proluminescent caspase-3/7 substrate (**b**). Flow cytometric analysis of PI uptake and PS exposure (**c**). LDH release (**d**). In **b** and **d**, graphs depict the mean ± SD of triplicate (**b**) or duplicate (**d**) cultures, and individual data values are plotted. In **c**, representative flow cytometry profiles and the graph that depicts percentages of Annexin V[+] cells (untreated, *n* = 4; AP20187-treated, *n* = 6) from four independent experiments are shown. Horizontal and vertical bars indicate the mean ± SD. Statistical significance was determined using Bonferroni's multiple comparisons test (**b–d**). ***$p < 0.001$. **e** *Gsdmd/Casp9*-DKO CL26-iCasp1 cells were transduced or not transduced with Casp9 or empty vector control. The cells were treated with 50 nM AP20187 for the indicated times. Western blot detection of cleaved caspases in cell lysates (**e**). In **a**, **b**, **d**, and **e**, data are from one representative of three independent experiments with similar results. Source data are provided as a Source Data file. (See also Supplementary Fig. 7)

of Bax/Bak, pro-apoptotic partners of Bid, significantly reduced the activation of caspase-3 (Supplementary Fig. 8c–f) without affecting tBid generation. These results indicate that Bid plays a critical role in the caspase-1-induced cell death through the intrinsic apoptosis pathway in the absence of GSDMD.

Asp 59 of Bid was previously shown to be the cleavage site of caspase-1, whereas Asp 75 was the site for granzyme B[42]. The reconstitution of *Gsdmd/Bid*-DKO CL26-iCasp1 cells with WT or the D75A mutant of Bid restored the tBid generation, caspase-3 activation, LDH release, and PS exposure in response to AP20187 treatment (Fig. 6e–g and Supplementary Fig. 9d). However, the D59A mutant of Bid failed to restore these responses, indicating that caspase-1 cleaves Bid at Asp 59 to induce apoptosis. tBid was

also produced in *Gsdmd*[−/−] BMMs during *S.* Typhimurium infection, while it was poorly produced in *Casp1*[−/−] BMMs (Fig. 7a). Knockdown of Bid suppressed the *S.* Typhimurium-induced caspase-3 activation (Fig. 7b), and diminished the apoptotic morphological changes in *Gsdmd*[-/-] BMMs (Fig. 7c). Consistently, *Gsdmd/Bid*-DKO BMMs showed reduced caspase-3 activation and LDH release compared to *Gsdmd*[−/−] BMMs during *S.* Typhimurium infection (Fig. 7d, e). *S.* Typhimurium infection and Val-boroPro treatment induced tBid production in *Gsdmd*-KO RAW264.7 cells, but not in *Gsdmd/Casp1*-DKO cells (Fig. 7f and Supplementary Fig. 10a). Caspase-3 activation and apoptotic morphological changes were induced in *Gsdmd*-KO RAW264.7 cells by *S.* Typhimurium and Val-boroPro, but were

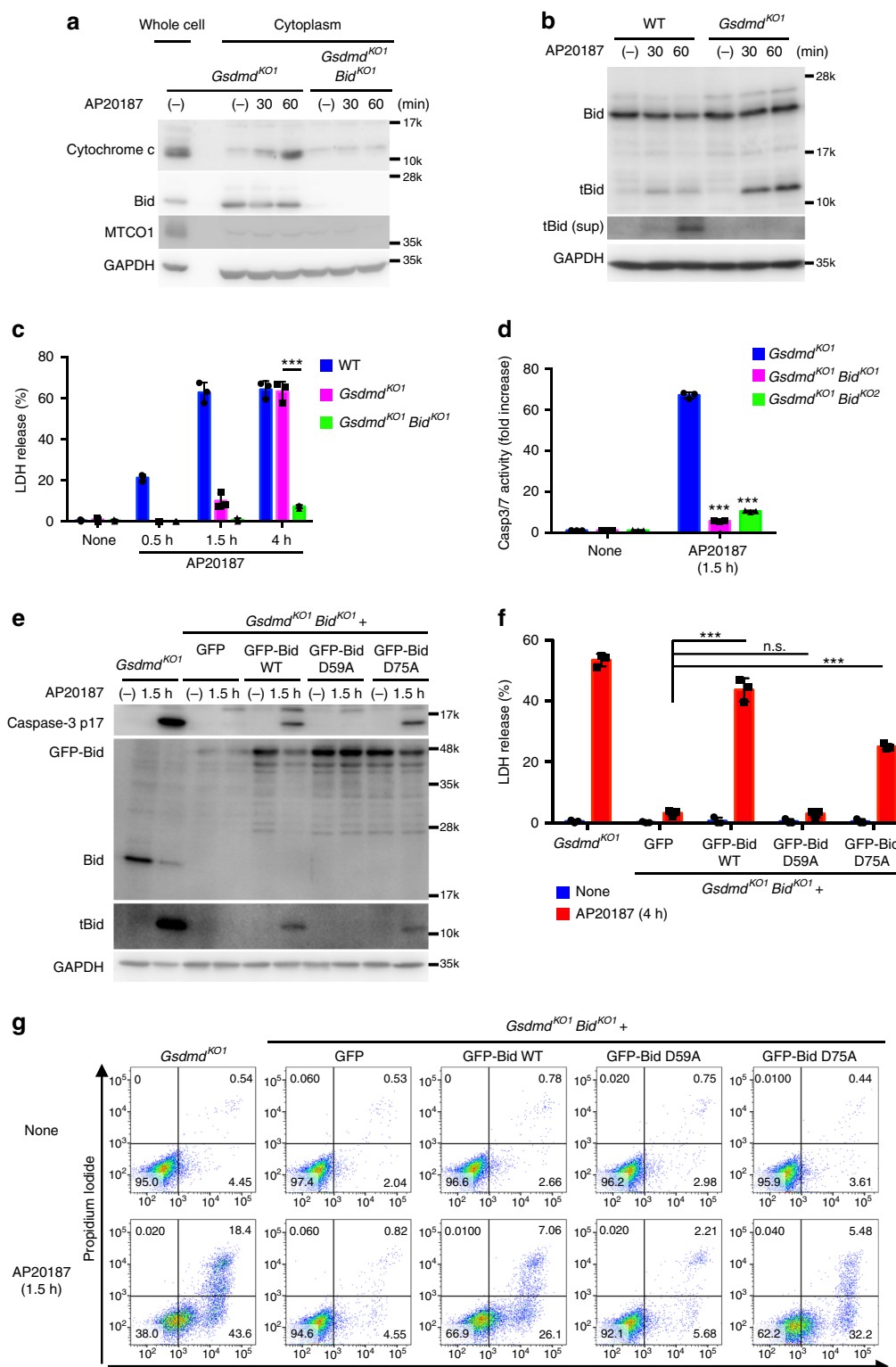

**Fig. 6** Bid plays a critical role in caspase-1-induced apoptosis. **a–d** WT, *Gsdmd*-KO or *Gsdmd/Bid*-DKO CL26-iCasp1 cells were treated with AP20187 for the indicated times. Cytochrome c, Bid, GAPDH (cytosolic marker) and MTCO1 (mitochondrial marker) in the whole cell lysates or cytosol fractions were detected by western blotting (**a**). Bid in the cell lysates and culture supernatants (sup) was detected by Western blotting (**b**). LDH release into culture supernatants (**c**). Caspase-3/7 activity in the cell lysates. **e–g** *Gsdmd*-KO CL26-iCasp1 cells and *Gsdmd/Bid*-DKO CL26-iCasp1 cells stably transfected with GFP, GFP-Bid, or GFP-Bid mutants were treated with AP20187 for the indicated times. Cleaved caspase-3 and Bid in cell lyesates were detected by western blotting (**e**). LDH release (**f**). Flow cytometric analysis of PI uptake and PS exposure (**g**). Percentages of Annexin V[+] cells are shown in Supplementary Fig. 9d. In **c**, **d**, and **f**, graphs depict the mean ± SD of triplicate cultures, and individual data values are plotted. Statistical significance was determined using the unpaired Bonferroni's multiple comparisons test. n.s., not significant; ***$p < 0.001$. Data are from one representative of three independent experiments with similar results (**a–g**). Source data are provided as a Source Data file. (See also Supplementary Figs. 8 and 9.)

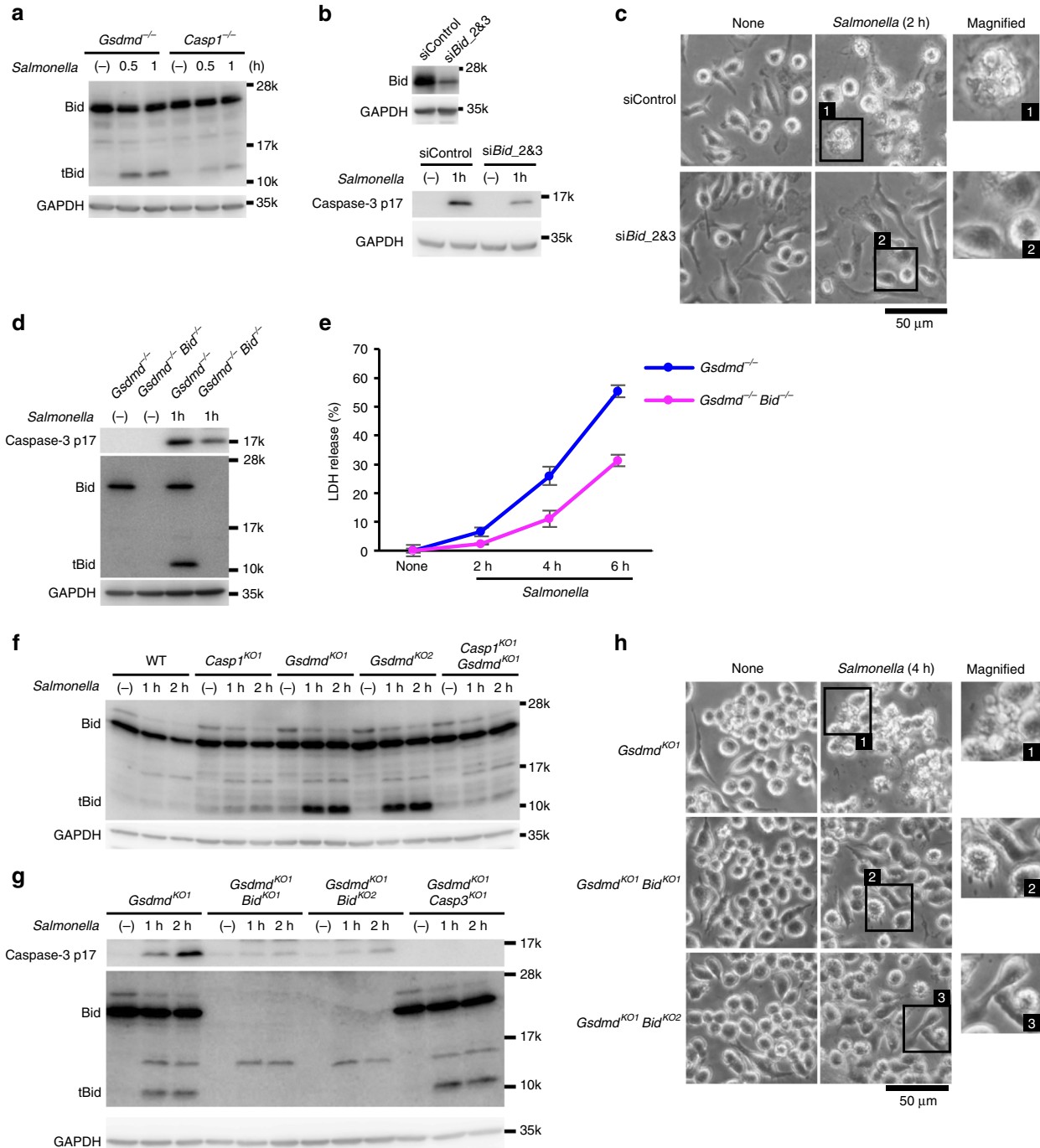

**Fig. 7** Bid is involved in caspase-1-induced apoptosis in macrophages. **a** BMMs from *Gsdmd⁻/⁻* and *Casp1⁻/⁻* mice were infected with *S.* Typhimurium as in Fig. 1d. Bid in cell lysates was detected by Western blotting. **b–e** *Gsdmd⁻/⁻* BMMs were transfected with control siRNA or *Bid* siRNAs (**b**, **c**). Two days after transfection, the cells were again transfected with the same siRNAs and incubated for an additional 2 days (**b**, **c**). BMMs were prepared from *Gsdmd⁻/⁻* and *Gsdmd⁻/⁻ Bid⁻/⁻* mice (**d**, **e**). The cells were then infected with *S.* Typhimurium as in Fig. 1d (**b**, **d**), Fig. 1c (**c**) or Fig. 1 b (**e**). Bid and cleaved caspase-3 in cell lysates were detected by western blotting (**b**, **d**). Microscopic images (**c**). LDH release (**e**). **f–h** RAW264.7 cells of the indicated genotypes were infected with *S.* Typhimurium as in Fig. 1f (**f**, **g**) or Fig. 1e (**h**). Western blot detection of cleaved caspase-3 and Bid in cell lysates (**f**, **g**). Microscopic images (**h**). In **e**, the graph depicts the mean ± SD of triplicate cultures. Data are from one representative of three biologically independent experiments (**a–e**) and three independent experiments (**f–h**) with similar results. Source data are provided as a Source Data file. (See also Supplementary Figs. 10 and 11.)

suppressed by the ablation of Bid (Fig. 7g, h and Supplementary Fig. 10b, c). In addition, the LDH release induced by Val-boroPro was significantly lower in *Gsdmd*/*Bid*-DKO RAW264.7 cells than in the *Gsdmd*-KO RAW264.7 cells (Supplementary Fig. 10d). From these results, we concluded that Bid is involved in the caspase-1-induced apoptosis in multiple cell types. The tBid production induced by caspase-1-activating agents was not affected by the depletion of caspase-3 or caspase-9 (Fig. 7g, Supplementary Fig. 9e, f and Supplementary Fig. 10b), indicating that Bid is upstream of these caspases.

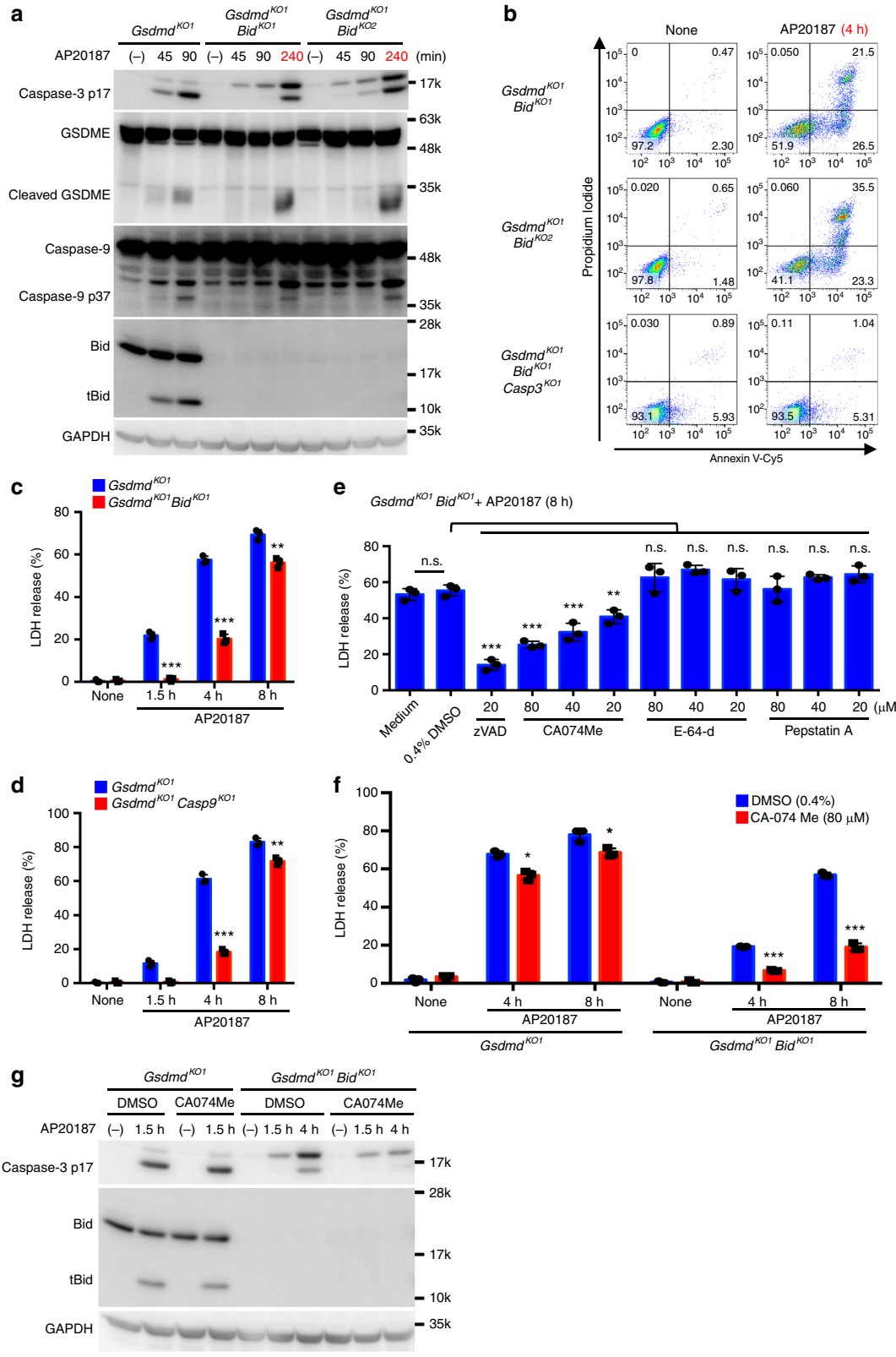

Then we tested the effect of caspase-1-induced apoptosis on the fate of *S*. Typhimurium in macrophages, the number of intracellular bacteria was determined by gentamicin protection assay (Supplementary Fig. 11). *S*. Typhimurium persisted in *Casp1*$^{-/-}$ BMMs more than in WT and *Gsdmd*$^{-/-}$ BMMs

(Supplementary Fig. 11a). *S*. Typhimurium could grow in *Gsdmd*/*Casp1*-DKO and *Gsdmd*/*Bid*-DKO RAW264.7 cells but not in WT and *Gsdmd*-KO RAW264.7 cells (Supplementary Fig. 11b). These results suggest that caspase-1-induced Bid-dependent apoptosis can contribute to the suppression of

**Fig. 8** A Bid-independent pathway is blocked by CA-074 Me. **a–c** *Gsdmd*-KO, *Gsdmd/Bid*-DKO, or *Gsdmd/Bid/Casp3*-triple KO CL26-iCasp1 cells were treated with 50 nM AP20187 for the indicated times. Cleaved caspases, GSDME, and Bid were detected by western blotting (**a**). Flow cytometric analysis of PI uptake and PS exposure (**b**). Percentages of Annexin V⁺ cells are shown in Supplementary Fig. 9c. LDH release (**c**). **d** *Gsdmd*-KO and *Gsdmd/Casp9*-DKO CL26-iCasp1 cells were treated with AP20187 for the indicated times, and LDH release was determined. **e** *Gsdmd/Bid*-DKO CL26-iCasp1 cells were pretreated with the indicated compounds for 1 h and then treated with AP20187 for 8 h. LDH release was determined. **f, g** *Gsdmd*-KO and *Gsdmd/Bid*-DKO CL26-iCasp1 cells pretreated with solvent control or CA-074 Me (80 μM) for 1 h were treated with AP20187 for the indicated times. LDH release was determined (**f**). Cleaved caspase-3 and Bid were detected by Western blotting (**g**). In **c–f**, graphs depict the mean ± SD of triplicate cultures, and individual data values are plotted. Statistical significance was determined using the unpaired Student's *t*-test (**c**, **d**, and **f**) or Bonferroni's multiple comparisons test (**e**). n.s., not significant; *$p < 0.05$, **$p < 0.01$, and ***$p < 0.001$. Data are from one representative of three independent experiments with similar results (**a–g**). Source data are provided as a Source Data file. (See also Supplementary Fig. 12.)

intracellular bacterial growth/survival in the absence of GSDMD.

**A Bid-independent pathway that is sensitive to CA-074 Me**. Although Bid played a critical role in the caspase-1-induced apoptosis and GSDME cleavage at early time points, *Gsdmd/Bid*-DKO CL26-iCasp1 cells showed caspase-3 and caspase-9 activation, GSDME cleavage, PS exposure, plasma membrane blebbing, and LDH release when the AP20187 treatment was prolonged (Fig. 8a–c and Supplementary Fig. 12a–c). Depletion of caspase-3 from the CL26-iCasp1 cells diminished GSDME cleavage, PS exposure, LDH release, and loss of viability (Fig. 8b and Supplementary Fig. 12b–d). These observations suggested that a Bid-independent pathway exists that transduces caspase-1 activation to apoptosis in a caspase-3-dependent manner. Similarly, the *Gsdmd/Casp9*-DKO CL26-iCasp1 cells died after an 8-h incubation with AP20187 (Fig. 8d). We found that Z-VAD-FMK, a pan-caspase inhibitor, and CA-074 Me, a cathepsin B inhibitor, suppressed the LDH release from *Gsdmd/Bid*-DKO CL26-iCasp1 cells treated with AP20187 for 8 h (Fig. 8e). CA-074 Me significantly reduced the LDH release and caspase-3 activation in *Gsdmd/Bid*-DKO cells, while it had no or only a marginal effect on these responses in *Gsdmd*-KO cells (Fig. 8f, g), suggesting that the inhibitory effect of CA-074 Me is specific for the Bid-independent pathway.

In Raw264.7 cells, unlike in CL26-iCasp1 cells, the involvement of the Bid-independent pathway in caspase-1-mediated apoptosis might be limited as *Gsdmd/Bid*-DKO Raw264.7 cells were highly protected from cell death even at 24 h after the Val-boroPro treatment (Supplementary Fig. 10d). Further experiments using *Gsdmd/Bid*-DKO mice are required to evaluate the importance of the Bid-independent pathway in macrophages.

**Caspase-1 induces apoptosis in naturally GSDMD null cells**. We next sought to ask whether caspase-1 would induce apoptosis in naturally GSDMD negative cells. To this end, we first employed L929 cells that do not express GSDMD (Supplementary Fig. 1). L929 cells transduced with Fv3-mCasp1 (L929-iCasp1-cells) underwent apoptosis accompanied with caspase-3 and Bid processing after AP20187 treatment (Fig. 9a–c). Ectopic expression of GSDMD changed the form of cell death in the L929-iCasp1 cells into pyroptosis (Fig. 9c–e), confirming that GSDMD determined the mode of caspase-1-induced cell death.

The above results suggest that activation of caspase-1 would induce apoptosis in caspase-1 positive but GSDMD negative or low cells under more physiologically or pathologically relevant conditions (Supplementary Fig. 13a). Analysis of a genome-wide expression dataset (GNF Mouse GeneAtlas V3) revealed that the GSDMD expression is high in macrophages and the small intestine in mice (Supplementary Fig. 13b). On the other hand, GSDMD is weakly expressed in the epidermis, mammary gland, mast cells, and marginal zone B cells, whereas caspase-1 is

expressed at high levels in the same samples (Supplementary Fig. 13b). Moreover, the GSDMD expression was very low in the spinal cord compared to the TEPMs, spleen, and RAW264.7 cells at the protein and mRNA levels, while the pro-caspase-1 and Bid proteins and the *Gsdme* gene transcript were detected in the same spinal cord specimens (Supplementary Fig. 13c–e). Thus, there are cell types that express caspase-1 without expressing substantial levels of GSDMD, in which caspase-1-induced apoptosis may occur. Moreover, primary cortical neurons have been demonstrated to undergo apoptosis accompanied with Bid cleavage in a caspase-1-dependent manner after oxygen/glucose deprivation (OGD)[28]. We found that GSDMD was not expressed in primary cortical neurons (Fig. 10a and Supplementary Fig. 13f). Consistent with the previous study, OGD induced the activation of caspase-3 and apoptosis accompanied with nuclear pyknosis and karyorrhexis in cortical neurons (Fig. 10b and Supplementary Fig. 13g). Furthermore, the OGD-induced apoptosis was diminished in the absence of caspase-1 or Bid (Fig. 10b). We also prepared bone marrow-derived mast cells (Fig. 10c). GSDMD mRNA levels were significantly lower in the cells than in BMMs (Fig. 10a). Stimulation with nigericin, an activator of the NLRP3 inflammasome, induced PS exposure and cell death in LPS-primed mast cells from WT mice, but not those lacking caspase-1 (Fig. 10d). Also, the activation of caspase-1 and caspase-3, tBid production, and GSDME maturation were induced during nigericin treatment, which are diminished in *Casp1*−/− mast cells (Fig. 10e). Mature GSDMD (the p30 fragment) was only faintly detected in nigericin-treated mast cells, while inactive p43 and p21 fragments of GSDMD were generated (Fig. 10f), suggesting that full-length GSDMD and GSDMD p30 were cleaved possibly by caspase-3 as previously described[32,39]. Besides, ablation of Bid significantly reduced caspase-3 activation and GSDME maturation induced by nigericin (Fig. 10g). These results support the idea that caspase-1-induced apoptosis occurs in cell types that express no or a little GSDMD. Accordingly, the type of cell death induced after caspase-1 activation should be carefully examined, as it could be pyroptosis or apoptosis, depending on the expression of GSDMD.

## Discussion

Taken together with previous studies[2,4,12,14,24,25], we propose a signaling network model for the cell death induced by inflammasomes (Supplementary Fig. 14). Our results indicate that caspase-1 can initiate apoptosis in the absence of GSDMD through Bid-dependent and Bid-independent pathways. The Bid-independent pathway was unmasked only in the absence of Bid. Early studies suggested that certain non-myeloid cells undergo caspase-1-dependent apoptotic cell death[26–29]. However, at present, caspase-1 is widely considered as an inflammatory caspase that initiates pyroptosis, probably because most of the recent studies on the inflammasome have been carried out using myeloid cells that abundantly express GSDMD. Thus, our current

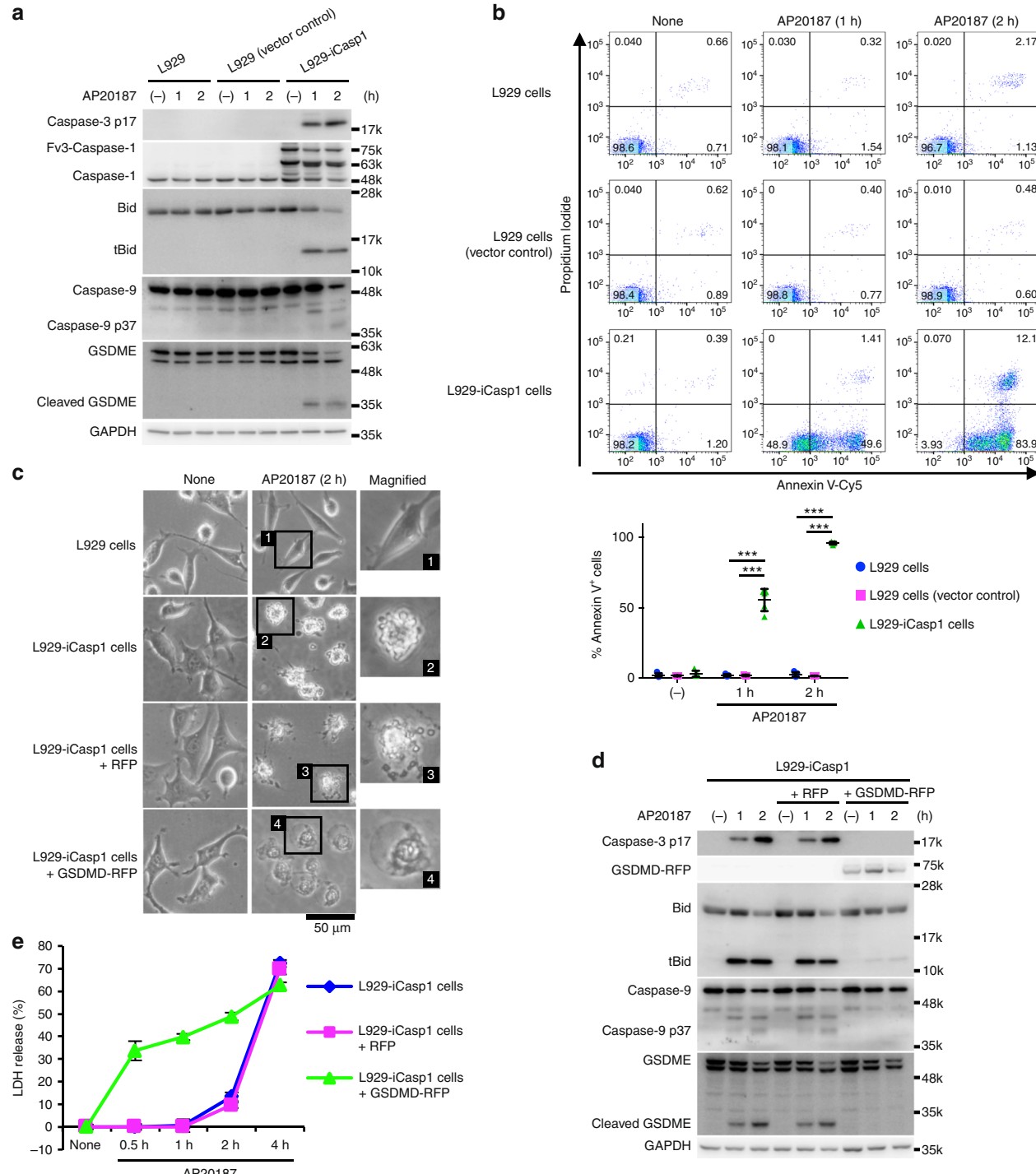

**Fig. 9** Caspase-1 induces apoptosis in L929 cells. **a, b** L929 cells, L929-vector control cells, and L929-iCasp1 cells were treated with 50 nM AP20187 for the indicated times. Caspases, Bid, and GSDME were detected by Western blotting (**a**). Flow cytometric analysis of PI uptake and PS exposure (**b**). Representative flow cytometry profiles and the graph that shows percentages of Annexin V+ cells (*n* = 5) from two independent experiments are shown. Horizontal and vertical bars indicate the mean ± SD. **c–e** L929-iCasp1 cells were transduced or not transduced with red fluorescent protein 657 (RFP) or GSDMD-RFP, and then treated with AP20187 for the indicated times. Microscopic images (**c**). Caspases, GSDMD, GSDME, and Bid were detected by Western blotting (**d**). LDH release (**e**). The graph depicts the mean ± SD of triplicate cultures. In **b** and **e**, statistical significance was determined using the Bonferroni's multiple comparisons test. ***p < 0.001. In **a** and **c–e**, data are from one representative three independent experiments with similar results. Source data are provided as a Source Data file. (See also Supplementary Fig. 13.)

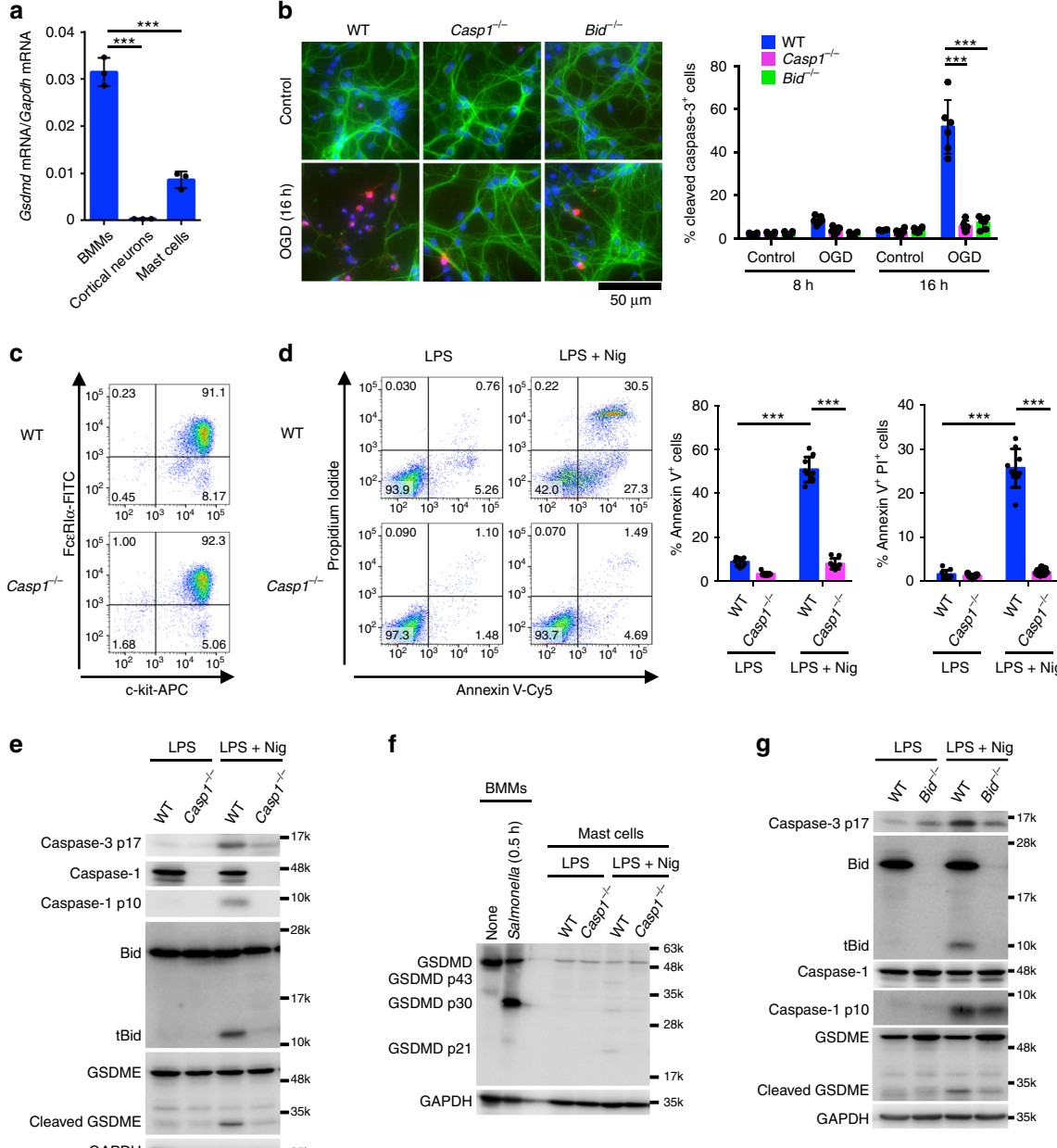

**Fig. 10** Caspase-1 induces apoptosis in primary cortical neurons and mast cells. **a** Quantitative RT-PCR analysis of GSDMD mRNA expression in BMMs, cortical neurons, and bone marrow-derived mast cells. GSDMD mRNA expression was calculated relative to GAPDH mRNA (the ΔCt method). Data are from three biologically independent experiments. **b** Primary cortical neurons were exposed to OGD for 8 and 16 h, and the cells were processed for immunofluorescent labeling of cleaved caspase-3 (red), β III-tubulin (green), and nuclei (blue). Representative fluorescent microscopic images from three biologically independent experiments are shown. The graph depicts percentages of cleaved caspase-3+ cells (six fields, >900 cells per group). **c** Surface expression of FcεRIα and c-kit in mast cells was determined by flow cytometry. **d–g** Mast cells of the indicated genotypes were primed with LPS (100 ng ml$^{-1}$) for 3 h and then stimulated with nigericin (10 μM) for 0.5 h (**e–g**) or 1 h (**d**). Flow cytometric analysis of PI uptake and PS exposure (**d**). Western blot detection of caspases, Bid, and GSDMs in cell lysates (**e–g**). In **c–g**, representative flow cytometry profiles (**c**, **d**) and Western blot images (**e–g**) from three biologically independent experiments are shown. In **d**, percentages of Annexin V+ cells and Annexin V+ PI+ cells (n = 9) are plotted. In **a**, **b**, and **d**, columns and bars represent the mean ± SD. Statistical significance was determined using the Bonferroni's multiple comparisons test (**a**, **b**, and **d**). ***$p < 0.001$. Source data are provided as a Source Data file

study provides an opportunity to revisit caspase-1-induced apoptosis and its pathophysiological functions.

Two reports on the role of caspase-1 in apoptotic signaling were recently published[32,43]. Taabazuing et al. demonstrated that Val-boroPro, a dipeptidyl peptidase (DPP) inhibitor, which induce a caspase-1-dependent cell death, induced apoptosis

accompanied by caspase-3 and caspase-7 activation in GSDMD-deficient RAW264.7 and THP-1 cells[32]. They also demonstrated that recombinant caspase-1 cleaved caspase-3 and caspase-7 in a cell free system. However, whether the cleaved caspases were functional and whether the direct cleavage really happened in cells were not examined. Sagulenko et al. showed that the

activation of the AIM2 inflammasome by DNA transfection induced caspase-3 activation even in caspase-8-deficient BMMs, and this caspase-3 activation was reduced when caspase-1 was knocked down[43]. These studies suggested that caspase-1 is involved in caspase-3 activation, although the mechanism of the pro-apoptotic action of caspase-1 remained unclear. Our results are consistent with these studies, and provide further conclusive evidence and molecular mechanisms for the caspase-1-induced apoptosis.

Not only caspase-1 but also caspase-8 is recruited to and activated in ASC specks in response to inflammasome activation[24,25]. This caspase-8 activation in ASC specks might have explained the inflammasome-induced apoptosis in GSDMD-deficient cells[24,25]. However, our results showed that the caspase-3 activation in response to inflammasome stimuli was higher in GSDMD-deficient than in caspase-1-deficient macrophages (Fig. 1d). He et al.[20] reported similar observations using ASC-reconstituted RAW264.7 cells. It is thus possible that caspase-1 cooperates with caspase-8 to trigger apoptosis after inflammasome formation in ASC-expressing GSDMD-deficient cells. Furthermore, the caspase-8 activation induced in *Gsdmd*-KO CL26-iCasp1 cells and *Gsdmd*-KO RAW264.7 cells by AP20187 and *S.* Typhimurium, respectively, was diminished by the ablation of caspase-3 (Fig. 4c, e). Therefore, caspase-8 activation can occur not only in ASC specks, but also downstream of the caspase-1-caspase-3 axis (Supplementary Fig. 14). This finding is supported by the observation that the inflammasome-induced caspase-8 activation was greater in GSDMD-deficient than in caspase-1-deficient macrophages (Fig. 1d)[20,44].

Although we demonstrated that Bid is required for caspase-1-initiated apoptosis, Bid is widely recognized as a caspase-8 substrate that mediates extrinsic apoptosis in type II cells[2,45,46]. In type I cells, caspase-8 directly activates effector caspases, whereas this mechanism is inefficient or not operational in type II cells. The expression of X-linked inhibitor of apoptosis protein (XIAP), an inhibitor of caspase-9 and effector caspases (caspase-3/7), in type II cells has been implicated in the latter phenomenon. Instead, in type II cells caspase-8-cleaved Bid (tBid) activates the intrinsic apoptosis pathway, in which mitochondria release proapoptotic proteins including cytochrome c, which activates caspase-9, and Smad, which relieves caspases from XIAP's inhibition. We previously reported that apoptosis mediated by the ASC-caspase-8 axis follows the type I/type II⁻rule. Importantly, purified active caspase-1 is reported to cleave procaspase-3 in vitro[47], although whether this occurs in cells needs to be confirmed. If caspase-1-mediated apoptosis also follows the type I/type II-rule, the requirement for Bid in this cell death pathway may differ according to cell type. Consistent with this possibility, CL26-iCasp1 cells are type II cells, as the TNF-induced apoptosis of the cells was Bid-dependent (Supplementary Fig. 15); however, it remains to be determined whether Bid is dispensable for the caspase-1-mediated apoptosis in type I cells.

The above discussion regarding the type I/type II rule may also be relevant to the Bid-independent pathway for caspase-1-initiated apoptosis in GSDMD-deficient cells, because (1) this Bid-independent pathway was inhibited by the cathepsin inhibitor CA-074Me (Fig. 8), and (2) cathepsins can play a role in apoptosis induction by degrading anti-apoptotic proteins including XIAP[48]. Therefore, a possible scenario is that in cells lacking both GSDMD and Bid, caspase-1 might directly or indirectly activate cathepsins, which in turn relieves caspases from XIAP's inhibition, and then caspase-1 might directly activate effector caspases to induce apoptosis even in type II cells. These assumptions need to be tested in future investigations. In addition, caspase-1 has also been suggested to activate caspase-6

and caspase-7[49,50], which were not required for caspase-1's induction of apoptosis in Bid-sufficient cells (Fig. 4f), but the involvement of these caspases in the Bid-independent pathway remains to be tested.

Our results suggest that GSDMD suppresses the caspase-1-induced caspase-3 activation in a manner dependent on its pore-forming activity (Fig. 2c). In particular, the caspase-3 activation was seen in *Gsdmd*-KO CL26-iCasp1 cells but not in WT CL26-iCasp1 cells 30 min after AP20187 treatment, at which time the WT cells were only partially lysed (Supplementary Fig. 3d, f). Therefore, it is unlikely that the caspase-3 activation was inhibited due to GSDMD-dependent cell disruption. We showed that ATP was released from GSDMD-expressing cells immediately after the activation of caspase-1 (Fig. 2g). ATP/dATP are theoretically small enough to pass through GSDMD-formed pores; thus, they are likely to diffuse through these pores. This could explain the attenuation of caspase-1-induced caspase-3 activation by GSDMD, because ATP/dATP are necessary for apoptosome formation[51,52]. In addition, tBid was also released after caspase-1 activation in a GSDMD-dependent manner (Fig. 6b). Strikingly, it has been demonstrated that active caspase-3 cleaves GSDMD to inactivate it[32,39]. Taken together, caspase-1-induced apoptosis and pyroptosis appear to suppress each other (Supplementary Fig. 14).

We propose that caspase-1-induced apoptosis occurs in cell types that do not or only weakly express GSDMD. Indeed, primary cortical neurons and mast cells underwent apoptosis in a caspase-1 and Bid-dependent manner (Fig. 10 and Supplementary Fig. 13). In mouse models of familial amyotrophic lateral sclerosis[53–55] and ischemia-induced brain injury[28,56], caspase-1 is reported to be involved in the disease pathogenesis, tBid generation, and caspase-3 activation in the spinal cord and the brain, respectively. Substantial amounts of caspase-1 and Bid, but no or little GSDMD, are expressed in the spinal cord (Supplementary Fig. 13). Accordingly, caspase-1-induced apoptosis is likely to participate in neurodegeneration under pathological conditions. Caspase-1-initiated apoptosis has also been implicated in UVB-induced apoptosis in human keratinocytes[57], cisplatin-induced apoptosis in the kidney[58], and hyperhomocysteinemia-induced endothelial apoptosis[59]. Interestingly, LPS-stimulated monocytes are reported to release microvesicles containing active caspase-1, which are taken up by different cells to deliver the cell death signal; for example, active caspase-1-containing microvesicles induce apoptosis in vascular smooth muscle cells and lymphocytes[60,61]. Hence, it may be speculated that caspase-1-induced apoptosis can occur in GSDMD-null cells in a non-cell-autonomous manner through the microvesicular transfer of active caspase-1 (Supplementary Fig. 13a). It is also reported that the 3 C protease of enterovirus 71 degrades GSDMD[62], raising the possibility that microbial pathogens that stimulate inflammasome formation and inactivate GSDMD simultaneously induce caspase-1-mediated apoptosis (Supplementary Fig. 13a). Collectively, it is possible that caspase-1 initiates apoptosis in a variety of cell types in many situations.

Caspase-1, which was identified as IL-1β-converting enzyme (ICE) in 1992–1993[63–65], was suggested to induce apoptosis for the first time in 1993[26]. The concept of pyroptosis was introduced in 2000[5]; this concept is now widely accepted, whereas caspase-1-induced apoptosis has not received much attention. However, our results clearly demonstrated that this caspase has the potential to initiate both pyroptosis and apoptosis. The role of apoptosis, which is currently believed to be a silent cell death, initiated by caspase-1, the inflammatory cytokine-converting enzyme, remains to be unveiled. We hope that this study, which may represent a turning point in the understanding of the effector mechanisms of the inflammasome, will promote research on the

pathophysiological roles of caspase-1 in different types of cells and tissues.

## Methods

**Mice**. Wild-type C57BL/6 mice were purchased from Japan SLC. The $Gsdmd^{-/-}$ mice were generated previously[66]. The $Casp1^{-/-}$ mice[67] carrying the 129-type nonfunctional $Casp11$ gene[10] and the $Pycard^{-/-}$ mice[68] were kindly provided by Dr. Winnie W. Wong (BASF Bioresearch Corporation) and Dr. Tetsuo Noda (Cancer Institute, Japanese Foundation for Cancer Research), respectively, and had been backcrossed to C57BL/6 mice. The $Bid^{-/-}$ mice[69] were kindly provided by Dr. Tetsuo Takehara (Osaka University Graduate School of Medicine). The $Gsdmd^{-/-}$ $Bid^{-/-}$ mice were generated by crossing $Gsdmd^{-/-}$ mice and $Bid^{-/-}$ mice. Mice were bred and maintained in pathogen-free animal facilities at Kanazawa University, and were used between 10–20 weeks of age. The spinal cord was collected from mice perfused with cold Hanks' balanced salt solution while anesthetized. The spleen, femurs, tibias, and peritoneal cells were harvested from mice after euthanasia. All animal experiments were approved by the Animal Care and Use Committee of Kanazawa University (AP-143305, AP-173853, AP-184013), and conducted in accordance with the Kanazawa University Animal Experimentation Regulations and the International Guiding Principles for Biomedical Research Involving Animals by the Council for International Organization of Medical Sciences and the international council for Laboratory animal science (December 2012).

**Reagents**. Val-boroPro (MedChemExpress, HY-13233A); AP20187 (MedChemExpress, HY-13992); Poly dA:dT (Sigma-Aldrich, P0883); Lipopolysaccharides from *Escherichia* (*E.*) *coli* K-235 (Sigma-Aldrich, L2018); z-VAD-fmk (R&D Systems, FMK001); recombinant mouse M-CSF (R&D Systems, 416-ML); Bacto-thioglycolate medium without dextrose (Difco, 0363-17-2); SUPERFASLIGAND Protein (Enzo Life Sciences, ALX-522-020-C005); Recombinant Murine TNF-α (PeproTech, 315-01A); nigericin (Cayman CHEMICAL, 11437); and Puromycin aminonucleoside (Focus Biomolecules, 10-2101) were purchased. YO-PRO-1 Iodide (Y3603), Blasticidin S (R21001), and Geneticin (11811023) were purchased from Thermo Fisher Scientific. CA-074 Me (4323-v), E-64-d (4321-v), and Pepstatin A (4397-v) were purchased from Peptide Institute (Osaka, Japan).

**Cell culture**. Colon-26 cells (purchased from the RIKEN BioResource Center), RAW264.7 cells (kindly provided by Dr. Kensuke Miyake, Institute of Medical Science, University of Tokyo), and L929 cells (purchased from Cell Resource Center for Biomedical Research, Institute of Development, Aging and Cancer, Tohoku University) were grown in RPMI 1640 (Sigma-Aldrich) supplemented with 10% fetal bovine serum (FBS), 100 U ml$^{-1}$ penicillin, and 100 μg ml$^{-1}$ streptomycin under a humidified atmosphere with 5% $CO_2$ at 37 °C. We confirmed that all the cell lines were free of mycoplasma contamination. Primary mouse bone marrow cells obtained from the femurs and tibias of 8–20-weeks-old mice were cultured in RPMI 1640 containing 10 ng ml$^{-1}$ M-CSF or 10% L929 conditioned medium for 7 days, and adherent cells were used as BMMs. Primary mouse bone marrow cells were cultured in RPMI 1640 containing 50% WEHI-3 conditioned medium for 28 days, and floating cells were used as bone marrow-derived mast cells. TEPMs were collected from the peritoneal cavity of 8–10-week-old mice 4 days after the intraperitoneal injection of a 3.0-ml volume of 3% thioglycolate medium. 293FT cells (Invitrogen) were cultured in DMEM (Sigma-Aldrich) supplemented with 10% FBS, 100 U ml$^{-1}$ penicillin, and 100 μg ml$^{-1}$ streptomycin.

**Primary cortical neuron culture and OGD**. Mouse primary cortical neurons were prepared from embryonic day 15 mice with Neuron Dissociation Solutions (FUJIFILM Wako Pure Chemical Corporation, 291-78101). Neurons were plated on poly-L-lysine-coated 24-well plate and cultured in Neurobasal Medium (Thermo Fisher Scientific, 21103049) containing 2% B27 supplement (Thermo Fisher Scientific, A3582801), 5% FBS, 0.5 mM L-Glutamine (WAKO, 076-00521), and 100 U ml$^{-1}$ penicillin and streptomycin at a density of $2 \times 10^5$ per cm$^2$. Cells were used for experiments on day 3 of culture. OGD on primary cortical neurons[28] was performed as follows. The culture medium was replaced by glucose-free Earle's balanced salt solution (Thermo Fisher Scientific, E2888), and the cells were placed in an anaerobic chamber with AnaeroPack (Mitsubishi Gas Chemical, A-04). Control cells were incubated in Earle's balanced salt solution with glucose in a normoxic incubator. After 8 or 16 h, cells were fixed and subjected to immunofluorescence staining with anti-cleaved Caspase3 (Cell Signaling Technology, #9661), Anti-Tubulin Antibody, β III isoform (Merck, MAB1637), and DAPI (Thermo Fisher Scientific, D1306).

**Induction of inflammasome formation**. *S.* Typhimurium (ATCC 14028) (obtained from American Type Culture Collection) was grown in LB broth at 37 °C overnight, diluted 1:50 into fresh LB medium, and further grown for an additional 2.5 h with vigorous shaking. The optical density at 600 nm of the culture was measured to determine the CFU. The bacteria were then diluted in RPMI 1640 without antibiotics for infection. BMMs and RAW264.7 cells cultured in antibiotic-free RPMI1640 medium at a density of $5 \times 10^4$ well$^{-1}$ in 96-well plates were infected with *S.* Typhimurium at an MOI of 5 and 40, respectively. The plates were centrifuged at $1500 \times g$ for 5 min and then transferred to a $CO_2$ incubator. After 1 h (BMMs) or 2 h (RAW264.7 cells), the cells were washed three times with antibiotic-free medium, and gentamicin (50 μg ml$^{-1}$) was added to the wells. TEPMs were transfected with Poly dA:dT using Lipofectamine LTX (Thermo Fisher Scientific).

**CRISPR/Cas9-mediated genome editing**. Guide RNA (gRNA) sequences were designed and are listed in Supplementary Table 1. Oligonucleotide pairs were phosphorylated, annealed, and ligated into the BbsI site of eSpCas9(1.1) (Addgene, 71814) or pX330 (Addgene, 42230). The plasmids were extracted from *E. coli* cells, and endotoxin was removed using 1% Triton-X 114 (Nacalai Tesque). The plasmids were transfected into CL26-iCasp1 cells and RAW264.7 cells using Lipofectamine 3000 and the Neon transfection system (Thermo Fisher Scientific), respectively. The cells were cloned by limiting dilution 5 days after transfection, and the editing of target genes was tested by western blotting or sequencing of the gRNA-target sites. KO clones made with gRNA_1 and gRNA_2 targeting each gene are indicated by superscript KO1 and KO2, respectively.

**Colon-26 and L929 cells expressing Fv3-caspases**. pC$_4$-Fv1E and pC$_4$M-Fv2E were obtained from ARIAD, and pC$_4$-Fv3E, which encodes three tandem FK506-binding protein Phe36Val (Fv) domains, was made by inserting the 654-bp XbaI-SpeI fragment from pC$_4$M-Fv2E into the SpeI site of pC$_4$-Fv1E. pC$_4$-Fv3E was digested by BamHI and SpeI, and the HA-tag sequence was replaced with an oligonucleotide linker containing the HA-tag sequence and the BamHI and SpeI restriction sites to generate pC$_4$-Fv3E-BS. PCR-amplified cDNAs of mouse caspase-1 and caspase-8 were cloned into pMOSBlue (GE Healthcare). The cDNA sequences excluding the stop codons were inserted into the BamHI and SpeI restriction sites of pC$_4$-FV3E-BS, and the resulting plasmids encoded pro-caspases fused with three N-terminal Fv domains and a C-terminal HA-tag, herein called Fv3-caspases. The Fv3-caspase-coding sequences were then transferred into the expression vector pEF-Bos. Colon-26 cells were co-transfected with each pEF-BOS- Fv3-caspase plasmid and a plasmid carrying the puromycin-resistance gene (pATM3-puro) using Lipofectamine 3000 (Thermo Fisher Scientific). The transfected cells were incubated with puromycin (10 μg ml$^{-1}$) for 1 week and cloned by limiting dilution, and the Fv3-caspase-expressing clones were then selected. To generate the expression vector pLenti-T2A-BFP, Cas9-coding sequences were deleted from and a HpaI site was inserted into pLentiCas9-T2A-BFP (Addgene, 78547) by PCR and ligation using In-Fusion HD Cloning Kit (Clontech, 639650). Fv3-caspase-1-coding sequences were amplified by PCR and inserted into the HpaI site of pLenti-T2A-BFP. Lentiviruses were produced in 293FT cells (Thermo Fisher Scientific) that were co-transfected with lentiviral packaging vectors (pMD2.G and psPAX2 from Addgene) along with pLenti-Fv3-caspase-1-T2A-BFP or pLenti-T2A-BFP by the calcium phosphate method. The culture supernatant including lentiviral particles was collected, passed through a 0.45-μm filter, and stocked at −80 °C. L929 cells were transduced with the lentiviruses and cultured in the presence of blasticidin (10 μg ml$^{-1}$) for 2 weeks to select for transduced cells, and the blue fluorescence-positive cells were isolated using a FACSAria II cell sorter (BD Biosciences).

**Small interfering RNA (siRNA)-mediated knockdown**. Bid was silenced using a MISSION esiRNA targeting mouse Bid (siBid_1; Sigma-Aldrich, EMU087401). The MISSION siRNA Universal Negative Control (Sigma-Aldrich, SIC-001) was used as a control siRNA for the siBid_1. Stealth RNAi siRNAs targeting mouse Gsdmd (MSS290579), mouse Bid (siBid_2 and siBid_3; MSS273392 and MSS273393, respectively), mouse Bax (MSS273360), and mouse Bak1 (MSS247040) were from Thermo Fisher Scientific. The Negative Universal control Med#3 (Thermo Fisher Scientific, 46-5373) was used as a control siRNA for the Stealth RNAi siRNAs. Cells were transfected with the siRNAs (50 nM) using Lipofectamine 3000 according to manufacturer's instructions. The cells were used in experiments 2 days after transfection. In some experiments, siRNA transfection was repeated twice, and the cells were used 4 days after the first transfection. The efficiency of knockdown was confirmed by western blotting.

**Separation of the cytoplasm fraction**. Cells were suspended in mitochondria isolation buffer containing 3 mM HEPES-KOH (pH 7.4), 210 mM mannitol, 70 mM sucrose, 0.2 mM EGTA, and Protease Inhibitor Cocktail (Nacalai Tesque, 25955-11), and then homogenized using the Pestle Grinder System (Thermo Fisher Scientific, 03-392-106) on ice. After centrifugation ($500 \times g$, 10 min, 4 °C), the supernatant was placed on an equal volume of 340 mM sucrose. Nuclei and unbroken cells were removed as the pellet by centrifugation at $500 \times g$, 10 min, 4 °C. The supernatant was transferred to a new tube and centrifuged at $10,000 \times g$, 10 min, 4 °C to remove mitochondria as the pellet. The supernatant was collected in another tube and used as the cytoplasm fraction.

**Caspase-3/7 assay and ATP measurement**. The caspase-3/7 activity in cell lysates was assayed using the Caspase-3/7-Glo luminescent kit (Promega, G8091). Cells were lysed in TBS containing 1% Triton-X 100 and 1 mM DTT, and 30 μl of the cell lysate was incubated with an equal volume of the reagent in a white 96-well plate for 5 min at room temperature. Luminescence values were measured with the

Tecan Infinite M200 plate reader equipped for luminescence detection. The ATP level in culture supernatants was measured using the Cell ATP assay reagent (Toyo-ink, 304-15361). The culture supernatants were heated at 95 °C for 10 min to inactivate ATPases immediately after collection, then 30 μl of supernatant was mixed with an equal volume of the reagent in a white 96-well plate. Luminescence was measured as described above. The amount of ATP was calculated by generating an ATP standard curve.

**LDH release assay**. The LDH released from cells into culture supernatants was measured using the CytoTox 96 Non-Radioactive Cytotoxicity Assay (Promega, G1780) according to the manufacturer's protocol. The percentage of LDH release was calculated as follows: % LDH release = (sample LDH activity-background LDH)(total LDH activity- background LDH)$^{-1}$ × 100.

**Flow cytometry**. Cells were stained with annexin V-Cy5 (BioVision, 1013; 1:1000 dilution) and propidium iodide (Immuno Chemistry Technologies, 638; 1:500 dilution) in annexin V binding buffer containing 10 mM HEPES (pH 7.4), 150 mM NaCl, 5 mM KCl, 1 mM MgCl$_2$, and 1.8 mM CaCl$_2$ for 15 min on ice. To assess mitochondrial membrane potential, cells were stained with 100 nM tetra-methylrhodamine methlester (TMRM; Thermo Fisher Scientific, T668) in culture medium for 15 min room temperature. Bone marrow-derived mast cells suspended in staining buffer (PBS, 1% BSA, 0.1% NaN$_3$) were pretreated with TruStain fcX (BioLegend, 101320) at 4 °C for 30 min and then stained with APC anti-mouse CD117 (BioLegend, 105811) and FITC anti-mouse FcεRIα (BioLegend, 134305) at 4 °C for 30 min. The cells were analyzed by flow cytometry using a FACSCanto II flow cytometer (BD Biosciences), and data were analyzed using FlowJo V10 software.

**Western blotting**. Cells were solubilized in SDS-Sample Buffer containing 20 mM DTT and boiled at 95 °C for 20 min. The spinal cord and TEPMs were lysed in TBS containing 1% Triton-X 100, 2 mM EDTA, and Protease Inhibitor Cocktail (Nacalai Tesque, 25955-11), and the protein concentration in the samples was assayed using the Pierce BCA Protein Assay Kit (23225) before solubilization in SDS-Sample Buffer. The samples were subjected to SDS–PAGE and transferred to PVDF membranes (Merck). The membranes were blocked with Blocking One (Nacalai Tesque) for 1 h at room temperature, and then incubated overnight at 4 °C with the following primary antibodies diluted in Immuno-enhancer Reagent A (Wako, 295-68614): cleaved Caspase3 (#9661, 1:1000), Caspase-3 (#9665, 1:1000), Caspase-6 (#9762, 1:500), Caspase-7 (#12827, 1:500), cleaved Caspase-8 (#8592, 1:500), Caspase-8 (#4790, 1:1000), Caspase-9 (#9504, 1:1000), Bid (#2003, 1:500), RIPK3 (#15828, 1:1000), and Cytochrome c (#4272, 1:500), Apaf-1 (#8969, 1:1000) from Cell Signaling Technology; MTCO1 (ab14705, 1:1000), GSDME (ab215191, 1:1000), GSDMD (ab209845, 1:1000), Caspase-1 (ab179515, 1:1000) from Abcam; Caspase-1 (sc-514, 1:1000) and GSDMD (sc-393581, 1:400) from Santa Cruz Biotechnology; Caspase-2 (MAB3507, 1:500) from Merck; and GAPDH (M171-3, 1:1000) from Medical & Biological Laboratories. The following secondary antibodies diluted in Immuno-enhancer Reagent B (Wako, 292-68624) were used: anti-rabbit IgG HRP-linked antibody (#7074, 1:1000), anti-mouse IgG HRP-linked antibody (#7076, 1:1000), and anti-rat IgG HRP-linked antibody (#7077, 1:1000) from Cell Signaling Technology. Most of Western blotting analyses were performed by an experimenter who was blinded to the samples. Uncropped scans of western blots are shown in Supplementary Fig. 16.

**Quantitative RT-PCR**. Total RNA was isolated from the spinal cord, spleen, and RAW264.7 cells using TRIzol Reagent (Thermo Fisher Scientific, 15596026) and reverse transcribed to produce cDNA with the SuperScript VILO cDNA Synthesis Kit (Thermo Fisher Scientific, 11754050) according to the manufacturer's instructions. Quantitative RT-PCR reactions were performed using the Express SYBR GreenER qPCR Super Mix (Thermo Fisher Scientific) and ViiA7 Real-Time PCR System (Thermo Fisher Scientific). The primers used for quantitative RT-PCR are as listed in Supplementary Table 1.

**Rescue experiments**. A multiple cloning site was inserted into the XbaI and BstBI sites of pLenti6/V5-DEST (Thermo Fisher Scientific) to generate pLenti6EB. The 778-bp EcoRI-XbaI fragment of pEGFP-N3 (Clontech) was inserted into the EcoRI and XbaI sites of pLenti6EB to generate pLenti6EB-GFP. The cDNA of mouse GSDMD was reverse transcribed from spleen RNA, amplified by RT-PCR, and cloned into pCR-Blunt (Thermo Fisher Scientific). The GSDMD I105N mutant was generated by site-directed mutagenesis. GSDMD-encoding sequences were amplified by PCR from the pCR-Blunt plasmids, digested by EcoRI, and inserted into the EcoRI site of pLenti6EB-GFP. Lentiviruses were produced in 293FT cells that were co-transfected with lentiviral packaging vectors (pLP1, pLP2, and pLP/VSV-G from Thermo Fisher Scientific) along with pLenti6EB-GSDMD-GFP or pLenti6EB-GSDMD I105N-GFP. *Gsdmd*-KO CL26-iCasp1 cells were transduced with the lentiviruses and cultured in the presence of blasticidin (2 μg ml$^{-1}$) for 2 weeks to select for transduced cells, and the EGFP-positive cells were isolated using a FACSAria II cell sorter. The cDNA of mouse Bid was obtained from Addgene (#3278), and mutations were introduced by site-directed mutagenesis. The plasmids were digested by EcoRI, and the Bid-encoding 624-bp DNA

fragments were inserted into the EcoRI site of pEGFP-C3 (Clontech). These plasmids were transfected into *Gsdmd/Bid*-DKO CL26-iCasp1 cells using Lipofectamine 3000. After selection with geneticin (500 μg ml$^{-1}$), the EGFP-positive cells were isolated using a FACSAria II cell sorter. Caspase-9 cDNA was amplified by PCR from mouse spleen cDNA and inserted into HpaI-digested pLenti-T2A-BFP using In-Fusion HD Cloning Kit. pLenti-Casp9-T2A-BFP was used to produce lentiviruses, and *Gsdmd/Casp9*-DKO CL26-iCasp1 cells were transduced with the viruses, as mentioned above. The GSDMD-encoding sequence except for the stop codon were amplified by PCR, digested by MluI, and inserted into the MluI site of pL-SFFV.Reporter.RFP657.PAC (Addgene, 61395). L929-iCasp1 cells were transduced with viruses produced in 293FT cells that were co-transfected with pL-SFFV.Reporter.RFP657.PAC-GSDMD, pMD2.G, and psPAX2. The transduced cells were selected in the presence of puromycin (2 μg ml$^{-1}$) for 2 weeks, and the red fluorescence-positive cells were isolated using a FACSAria II cell sorter.

**SCAT3-expressing cells**. *Gsdmd*-KO CL26-iCasp1 cells were transfected with pCDNA3.1-SCAT3[37] using Lipofectamine 3000. After selection with geneticin (500 μg ml$^{-1}$), the Venus-positive cells were isolated using a FACSAria II cell sorter.

**Microscopy and time-lapse imaging**. Light micrographs of cells were captured using a digital camera (Olympus, DP21) mounted on an inverted microscope (Olympus, CKX41). Time-lapse imaging was performed using a fluorescence microscope (Keyence, BZ-9000). Cells were treated with 50 nM AP20187 in the presence or absence of Yo-Pro-1 (1:1000 dilution) at 37 °C in a humidified atmosphere of 5% CO$_2$ in air. The YO-PRO-1, CFP, and CFP-Venus FRET images were acquired through a GFP filter set (Keyence, OP-87763), a CFP filter set (Chroma, 49001), and a CFP-YFP FRET filter set (Chroma, 49052), respectively. Microscopic images of cells were randomly selected from sample cultures, while magnified images of representative cells with normal, necrotic or apoptotic morphology are also shown.

**Immunoprecipitation**. CL26-iCasp1 cells were plated at a density of $10^6$ cells per 10 cm plate and incubated overnight. The cells were treated with or without 50 nM of AP20187 for 1 h, washed with PBS twice, and lysed in 0.5 ml lysis buffer (0.5% Nonidet P-40, 50 mM HEPES pH7.4, 150 mM NaCl, 10% glycerol, 2 mM EDTA, 1 mM DTT) containing Protease Inhibitor Cocktail and Phosphatase Inhibitor Cocktail (Nacalai Tesque, 03969-21) on ice for 10 min. The cell lysates were cleared by centrifugation for 15 min at 20000 × g at 4 °C and then reacted with anti-caspase-9 antibody (CST, #9504, 1:150) for 2 h at 4 °C with gentle rotation. Protein G Sepharose beads (GE Healthcare, 17-0618-02) were washed with three times lysis buffer and added to the cell lysate and antibody mixture (20 μl of 50% slurry per plate). The mixture was gently rotated for 2 h at 4 °C, and the beads were centrifuged and washed five times with lysis buffer. The immunoprecipitated proteins were eluted by adding 100 μl SDS-Sample Buffer containing 20 mM DTT and heating at 95 °C for 5 min.

**WST-1 assay**. Cell viability was determined by a colorimetric assay using 2-(4-iodophenyl)-3-(4-nitrophenyl)-5-(2, 4-disulfophenyl)-2H-tetrazolium mono-sodium salt (WST-1; Dojindo, W201) and 1-methoxy-5-methylphenazinium methylsulfate (1-Methoxy PMS; Dojindo, M003). WST-1 and 1-Methoxy PMS were dissolved in PBS at 10 mM and 0.4 mM, respectively. Equal volumes of the solutions were mixed, and the mixture was added to cell cultures (10 μl per 100 μl culture medium). The cells were incubated for 2 h, culture supernatants were transferred to a 96-well plate, and the net absorbance at 450 nm was measured using a microplate reader.

**Gentamicin protection assay**. Macrophages were infected with *S.* Typhimurium and treated with gentamicin as mentioned above. The cells were washed with PBS and lysed in 0.1X PBS/0.05% Triton X-100. The cell lysates were serially diluted with PBS/1% glycerol and plated on LB agar. Plates were incubated at 37 °C overnight, and colonies were counted.

**Web resources**. The URLs for web resources used in this study are as follows: BioGPS, http://biogps.org/#goto=welcome; GNF Mouse GeneAtlas V3 (GeneAtlas MOE430, gcrma), https://www.ncbi.nlm.nih.gov/geo/query/acc.cgi?acc=GSE10246.[70]

**Statistics**. Sample size was determined according to our experience and literature reporting similar experiments. Normality and variance were not examined, because sample size is too small to have sufficient power. Data were analyzed by a two-tailed unpaired *t*-test or two-way analysis of variance with Bonferroni's post-test, depending on the number of groups being compared. GraphPad Prism software 6.0 was used for the statistical analyses. A *p*-value less than 0.05 was considered to be statistically significant.

## Data availability

All data supporting the findings of this study are available within the article and its Supplementary Information files or are available from the corresponding authors upon reasonable request. The source data underlying Figs. 1a, b, 2b, d, g, 3b, f, 4g, i, 5b–d, 6c, d, f, 7e, 8c–f, 9b, e, 10a, b, d, and Supplementary Figs. 2c, 3b, c, e, f, 4b, d–g, 5d–f, 6a, b, 7a, b, 8a, c, d, f, 9b–d, 10d, 11a, b, 12c, d, 13d, e, and 15 are provided as a Source Data file. Uncropped scans of western blots are shown in Supplementary Fig. 16.

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

## Acknowledgements

This work was supported by Grant-in-Aid for Scientific Research on Innovative Areas (KAKENHI Grant Number 26110002) and by Grant-in-Aid for Challenging Exploratory Research (KAKENHI Grant Number 15K15078) from the Japan Society for the Promotion of Science (to T. Suda). This work was also supported by the Japan Society for the Promotion of Science (KAKENHI Grant Number 26460523) and the Institute for Frontier Science Initiative (InFiniti), Kanazawa University (to K.T.).

## Author contributions

K.T. and T. Suda designed this project. K.T. performed most of the experiments and wrote the manuscript. S.N., S.H., and M.R.M. helped with the experiments, including flow cytometry, western blotting, and preparation of bone marrow cells. O.H., D.T.N., T. H., and T.M.L. performed OGD experiments and prepared spinal cord samples. Y.Y. and M.M. provided the SCAT3 and helped with the FRET imaging. T.K., H.K., and M.S. prepared cells and mice used in this study. T. Shiroishi provided the *Gsdmd*$^{-/-}$ mice. T. Suda supervised the study and edited the manuscript. All authors reviewed the results and approved the final version of the manuscript.

## Additional information

**Competing interests:** The authors declare no competing interests.

