## [Peer Review File · Nature Communications]

Reviewers' comments:

Reviewer #1 (Remarks to the Author):

This manuscript reports how caspase-1 can activate Bid to trigger a caspase-9/caspase-3 dependent cell death in the absence of GSDMD. The authors also show that caspase-1 induced death can occur in the absence of Bid at later time points, and provide some evidence to suggest that this may result from cathepsin activity. The findings made are novel and are based on elegant experiments and rigorous genetic data. Moreover, they highlight how a physiological cell death might occur in caspase-1 expressing cells where GSDMD levels are low or absent. I am also confident that this manuscript will generate significant interest in the field as they significantly expand upon, and provide important corrections to, the current model whereby caspase-1 has been proposed to activate caspase-3 directly to induce apoptosis. I have several suggestions for the authors that I hope may improve and solidify the conclusions made:

1. Figure 3. In the absence of GSDMD, the author's data suggest that Caspase-1 activation of caspase-3 results in PS exposure and death. They show that caspase-1/-3 activation of GSDME is required for death, but not PS exposure. I am surprised the authors claim that GSDMD/GSDME deficient cells are therefore apoptotic, as PS externalization does not directly demonstrate a loss of viability. Therefore more evidence that these cells are undergoing apoptotic death is required. For example, have the cells stopped responding to external stimulus (e.g. TNF stimulation of NF- κ B) and/or is intracellular ATP levels/metabolism abolished? At later time points, after 4 hr, does caspase-1 eventually induce membrane rupture (i.e. allow propidium iodide uptake or cause LDH release) when GSDMD and GSDME are deleted, as this would provide the strongest evidence that it is apoptotic death being observed? If so, is this death entirely dependent on apoptotic caspases, caspase-3 and -7, or is it blocked by lysosomal/cathepsin protease inhibition (e.g. CA074)? If the later holds true, can the death really be called apoptosis?
2. Along similar lines to point 1, at the later time points, are caspase-3 and/or GSDME important for GSDMD/Bid-independent caspase-1 killing? i.e. could caspase-1 cleave caspase-3 to activate GSDME and induce cell death in the absence of GSDMD/Bid? This should be tested as the cell death modality should then be classified as pyroptosis, not apoptosis. Although the authors implicate cathepsin activity, the use of up to 80 μ M CA074 is extremely high, which may result in off-target effects (for example, can the authors perform additional control experiments to show that CA074 does not inhibit caspase-1/Bid-dependent cell death and/or examine if loss of lysosomal function, such as treatment with Bafilomycin A1, limits Bid-independent killing).
3. Please include molecular weight markers on all western blots to help readers gauge their accuracy.

Reviewer #2 (Remarks to the Author):

GSDMD is the recently identified executor of pyroptosis. It has however been known that GSDMD/-BMDMs nevertheless undergo a cell death program, which involves the activation of apoptotic caspases by caspase-1 and the induction of late necrosis (PMID: 26611636, 29281832). How exactly caspase-1 activates apoptotic caspases and executes this apoptotic program was unknown, although some reports indicated that caspase-1 might directly cleave caspase-3 (PMID: 28392147). Here the authors however show that Caspase-1 induces apoptosis (and late necrosis) in absence of GSDMD, and that this not only involves the activation of caspase-3, but all apoptotic caspases. Critical appears to be the activation of Caspase-9 and the formation of the apoptosome complex, since in absence of Caspase-9, Casp-3 is no longer processed. Their data suggest that Caspase-1 cleaves Bid, which then drives cytochrome C release and apoptosome assembly. They furthermore show that even in GSDMD/Bid dKO another pathway is activated, which causes apoptosis.

This is a solid study that investigates a timely topic, ie. the nature of GSDMD-independent cell death observed after caspase-1 activation. The conclusions are for the most part justified and the paper will be an interesting manuscript for Nat Comms with some additional experiment, in particular when verifying the results in a more physiological setting. To improve the manuscript, I would suggest to address the following points:

1. The authors should provide additional data to support that caspase-1 cleaves Bid to induce mitochondrial outer membrane permeabilization and apoptosome assembly, such as showing mitochondria membrane depolarization and Apaf-1/Casp-9 oligomerization. They should also demonstrate that this involves Bax/Bak.
2. The microscopy is poor. Often the morphological changes, that the authors refer to, are not visible in the figures. I would suggest to repeat the imaging using a better microscope, and to include galleries and close-ups of individual cells that clearly show the respective morphology.
3. Most of the study has been done with CL26 cells expressing chemically-activatable caspase-1. The CL26 cells are a colorectal carcinoma cells, and neither used by other researchers in the field nor very physiological. This casts some doubt on the physiological significance of the findings. The results need to be verified in mouse BMDMs, which is the only universally accepted model cell line for murine inflammasome activation. Furthermore, they should show the expression levels of all inflammasome and apoptotic signaling components and compare this to levels found in BMDMs (such as GSDMD, GSDME, ASC and others).
4. I am very skeptical about the proposed involvement of GSDME in causing late necrosis after activation of Caspase-1 in GSDMD^{-/-} cells. Although I agree with that GSDME can be cleaved and activated by apoptotic caspases, recent data published by N. Kayagaki suggest that GSDME is not involved in causing late necrosis (PMID: 29491424). Our own results support this notion, and it appears that BMDMs express too little GSDME, so that even if it is cleaved as shown by Lee et al. (PMID: 29491424), it will not be able to induce necrosis. Consistently, Feng Shao's group has reported, that only cell lines with high GSDME expression levels can induce GSDME-dependent necrosis (PMID: 28459430). In this regard it would be essential to know if CL26 cells express high GSDME levels. Furthermore, it is crucial for the conclusions that the authors want to take (see Figure 3f) to test BMDMs from GSDMD/GSDME dKO mice.
5. The authors should analyze either combined supernatant+lysates samples, or show the supernatant samples in addition to the cell lysates. In many of their blots, such as 1d, Casp-1 p10 or Casp3 p17 is not seen in WT cells. Presumably it is made, but not visible since it is released into the supernatant upon cell lysis.
6. GSDME cleavage blots will need to be shown in most of the figures in addition to the caspase and bid cleavage blots.
7. As for Figure 4: How do are the different caspases (Casp-2 and Casp-8) activated in the context of AP20187 treatment?

Minor comments:

1. Some of the blots need better loading controls, such as 1e and others.
2. Verify the specificity of the Bid inhibitor.
3. Taabazuing et al. have shown that Caspase-1 can directly cleave and activate Caspase-3, this should be more clearly discussed in the text.

Reviewer #3 (Remarks to the Author):

Title: Caspase-1 initiates apoptosis in the absence of gasdermin D

Summary/Major Criticism: In this manuscript, the authors show that in the presence of inflammasome stimuli and the absence of gasdermin D, caspase-1 leads to the activation of classical markers of apoptosis. Most notably, caspase-1 leads to caspase-3 cleavage via the activation of Bid/caspase-9. This leads to apoptosis as well as secondary necrosis mediated by GSDME. The authors' data also suggest caspase-1 activation in the absence of GSDMD leads to a Bid-independent apoptosis pathway, which remains to be elucidated. Although this manuscript takes a detailed mechanistic approach to dissect caspase-1-induced apoptosis, the authors mainly use a highly engineered cell line and the relevancy of this pathway is not demonstrated. Given that other recent publications (Lee et al., 2018, *Scientific Reports*, 8:3788 and Schneider et al., 2017, *Cell Reports* 21, 3846-3859) have described caspase-1-mediated apoptosis without identifying a physiologically relevant cell type/disease context, it is unclear how this study expands on these previous results. The authors even identify cells where this cell death pathway may occur but do not go on to test whether caspase-1-mediated apoptosis actually occurs in these cells. Thus, a major weakness of this manuscript is the lack of evidence of physiological significance. If the authors were to demonstrate that caspase-1-mediated apoptosis occurs via Bid/caspase-9/caspase-3 in a "wild-type" cell type, this would demonstrate relevancy and significance of the cell death pathway they describe.

Major comments:

1. For the in vitro *Salmonella* infection assays, can the authors please explain why they used streptomycin/penicillin to kill *Salmonella* after infection instead of the standard gentamicin? Additionally, the pictures in Fig. 1c,f, 3e, and 7c,f appear to show high concentrations of extracellular *Salmonella*. Standard in vitro *Salmonella* infection assays involve washing the cells to remove extracellular bacteria before adding on fresh media with antibiotics to maintain only intracellular infection. Can the authors please clarify if they performed a wash step? High amounts of extracellular bacteria can contribute to LDH signal, making removal of excess extracellular bacteria critical for LDH assays. If the authors did not follow standard gentamicin protection assays, the authors should repeat the *Salmonella* infection experiments using this standard method to ensure their current results correlate with the field standard for this type of assay.
2. For the in vitro *Salmonella* assays, did the authors quantify levels of intracellular bacteria? Inflammasome activation and pyroptosis are important innate defense mechanisms that restrict intracellular pathogens. Providing CFU counts/bacterial replication assay data at different time points for the intracellular *Salmonella* infections may help the authors demonstrate when the different pathways they have identified play a role in a physiologically relevant scenario.
3. For the caspase-9 experiments, did the authors perform complementation of the caspase-9 knockout cells to show restoration of the cell death phenotype? If so, these data should be included in the supplemental figures.
4. The authors frequently reference LDH release as evidence of apoptosis. However, LDH is released during pyroptotic and necroptotic cell death, not apoptotic cell death. How do the authors explain using LDH as a measure of apoptotic cell death? When the authors knock out Gsdme, they see reduced levels of LDH release. Since GSDME has been shown to mediate secondary necrosis, these data suggest the authors are observing secondary necrosis as measured by LDH release as well as apoptosis as measured by phosphatidylserine exposure simultaneously.
5. The authors state that previous studies showed caspase-3 can cleave GSDME. Given that the authors showed GSDME mediates LDH release and secondary necrosis in their system, did the authors look for GSDME cleavage in the presence/absence of caspase-3, caspase-9, and Bid? These data would strengthen their argument and further elucidate this pathway.
6. Based off the data in Figure 8, the authors suggest there is a Bid-independent apoptosis pathway that occurs at later time points (i.e., 8 hr instead of ≤ 4 hr). However, this conclusion is mostly based on LDH data, which is indicative of pyroptosis/necroptosis. Given that the authors looked at the contribution of GSDME in their system in a previous figure, can the authors please explain why they

did not examine the role of GSDME in this Bid-independent pathway?

Minor comments:

1. The authors should recheck the manuscript for spelling and grammar errors (e.g., Salmonella spelled Salomonella or caspase spelled capase or caspas).
2. Page 13, line 23, "Ding et al. reported..." This citation reference is wrong. The paper the authors are referencing is He et al. and is listed as reference number 20 in their references list.
3. In the Results section titled "Caspase-1-induced apoptosis involves caspase-9," lines 21 and 24, the authors incorrectly refer to Fig. 4d, e and then Fig. 4f, g when they should refer to Fig. 4e, f and Fig. 4g, h, respectively.
4. In the legend for Fig. 5a, the authors state that the Western blot probes for Bid. However, the provided image does not indicate Bid was examined. The authors should correct the legend or blot image accordingly.

Point-by-point reply to the reviewers' comments

Reviewer #1:

This manuscript reports how caspase-1 can activate Bid to trigger a caspase-9/caspase-3 dependent cell death in the absence of GSDMD. The authors also show that caspase-1 induced death can occur in the absence of Bid at later time points, and provide some evidence to suggest that this may result from cathepsin activity. The findings made are novel and are based on elegant experiments and rigorous genetic data. Moreover, they highlight how a physiological cell death might occur in caspase-1 expressing cells where GSDMD levels are low or absent. I am also confident that this manuscript will generate significant interest in the field as they significantly expand upon, and provide important corrections to, the current model whereby caspase-1 has been proposed to activate caspase-3 directly to induce apoptosis. I have several suggestions for the authors that I hope may improve and solidify the conclusions made:

Comment #1. Figure 3. In the absence of GSDMD, the author's data suggest that Caspase-1 activation of caspase-3 results in PS exposure and death. They show that caspase-1/-3 activation of GSDME is required for death, but not PS exposure. I am surprised the authors claim that GSDMD/GSDME deficient cells are therefore apoptotic, as PS externalization does not directly demonstrate a loss of viability. Therefore more evidence that these cells are undergoing apoptotic death is required. For example, have the cells stopped responding to external stimulus (e.g. TNF stimulation of NF-kB) and/or is intracellular ATP levels/metabolism abolished? At later time points, after 4 hr, does caspase-1 eventually induce membrane rupture (i.e. allow propidium iodide uptake or cause LDH release) when GSDMD and GSDME are deleted, as this would provide the strongest evidence that it is apoptotic death being observed? If so, is this death entirely dependent on apoptotic caspases, caspase-3 and -7, or is it blocked by lysosomal/cathepsin protease inhibition (e.g. CA074)? If the later holds true, can the death really be called apoptosis?

Answer to comment #1:

First of all, authors appreciate the reviewer for his/her positive rating of our manuscript and for valuable suggestions to improve our manuscript.

Actually, we did not literally describe that *Gsdmd/Gsdme*-double KO (DKO) cells were apoptotic (or died by apoptosis) in our manuscript. We just described that GSDME mediates secondary necrosis/pyroptosis in the caspase-1-initiated apoptosis of GSDMD-deficient CL26-iCasp1 cells. However, it is true that we thought that they underwent apoptotic cell death after AP20187 treatment, because in addition to PS externalization, they exhibited extensive membrane blebbing and cell fragmentation (Supplementary Fig. 5c that corresponds to original Supplementary Fig. 3f). To confirm further this notion, we performed several new experiments, and demonstrated caspase-3 processing (new Supplementary Fig. 5b), loss of viability (assessed by WST-1 assay, new Supplementary Fig. 5f) and closer views of deadly morphology (new Supplementary Fig. 5g) of *Gsdmd/Gsdme*-DKO cells after AP20187 treatment. In addition, *Gsdmd/Casp3*-DKO cells exhibited better viability (WST-1 assay) and healthier morphology than *Gsdmd/Gsdme*-DKO at 24 h after AP20187 treatment, suggesting that the AP20187-induced cell death of *Gsdmd/Gsdme*-DKO cells was largely caspase-3 dependent (new Supplementary Fig. 5g).

The reviewer also raised a question whether caspase-1 eventually induce membrane rupture at later time points, when GSDMD and GSDME are deleted. As shown in our new Supplementary Fig. 5e, LDH release from *Gsdmd/Gsdme*-DKO CL26-iCasp1 cells, which was about 10% at 4 h after AP20187 treatment, increased to about 65% at 24 h. In contrast, LDH release from *Gsdmd/Casp3*-DKO CL26-iCasp1 cells remained about 10% even at 24 h.

Together with our original data, these new results indicate that although GSDME is responsible for the rapid secondary necrosis following caspase-1-initiated apoptosis of *Gsdmd*-KO CL26-iCasp1 cells, other mechanisms (either active and/or passive ones) can induce delayed secondary necrosis in *Gsdmd/Gsdme*-DKO cells under the same conditions.

These results are now described in the new manuscript as follows:
(Page 8, line 21 - Page 9, line 6) We found that the KO of GSDME in *Gsdmd*-KO CL26-iCasp1 cells severely reduced the cell membrane damage observed within 4 h after AP20187 treatment, as determined by LDH release and staining with the cell-impermeant fluorescent dyes propidium iodide and YO-PRO-1 (Fig. 3f, g; Supplementary Fig. 5a-d; Supplementary Movie 3), while GSDME ablation did not affect the PS exposure, membrane blebbing, or caspase-3 activation in the CL26-iCasp1 cells (Fig. 3g and Supplementary Fig. 5b-d). However, *Gsdmd/Gsdme*-DKO CL26-iCasp1 cells became necrotic, and extensive LDH release was observed by 24 h after AP20187 treatment (Supplementary Fig. 5e-g). These results indicated that GSDME mediates a rapid secondary necrosis/pyroptosis after the caspase-1-induced apoptosis, but is not essential for the cell lysis at later times.

Comment #2. Along similar lines to point 1, at the later time points, are caspase-3 and/or GSDME important for GSDMD/Bid-independent caspase-1 killing? i.e. could caspase-1 cleave caspase-3 to activate GSDME and induce cell death in the absence of GSDMD/Bid? This should be tested as the cell death modality should then be classified as pyroptosis, not apoptosis. Although the authors implicate cathepsin activity, the use of up to 80 μ M CA074 is extremely high, which may result in off-target effects (for example, can the authors perform additional control experiments to show that CA074 does not inhibit caspase-1/Bid-dependent cell death and/or examine if loss of lysosomal function, such as treatment with Bafilomycin A1, limits Bid-independent killing).

Answer to comment #2:

To investigate whether GSDMD/Bid-independent caspase-1 killing is caspase-3-dependent, we newly prepared *Gsdmd/Bid/Casp3*-triple KO CL26-iCasp1 cells. *Gsdmd/Bid/Casp3*-triple KO cells survived at least until 24 h after caspase-1 activation (new Supplementary Fig. 12b, c, d), suggesting that caspase-3 is required for the Bid-independent pathway of caspase-1-induced cell death in GSDMD-null cells. In addition, GSDME was cleaved in *Gsdmd/Bid*-DKO CL26-iCasp1 cells treated with AP20187 in a caspase-3-dependent manner, which suggests that caspase-3 cleavage of GSDME led to cell lysis in the absence of GSDMD and Bid. However, *Gsdmd/Bid*-DKO CL26-iCasp1 showed PS externalization and plasma membrane blebbing, which are typical of cells undergoing apoptosis, before cell lysis (new Fig. 8b, and Supplementary Fig. 12a). Moreover, these apoptotic changes were diminished in the absence of caspase-3 (new Fig. 8b and Supplementary Fig. 9c). Therefore, we think that *Gsdmd/Bid*-independent cell death induced by caspase-1 is classified as apoptosis.

These results are now described in the new manuscript as follows:
(Page 12, lines 16-23) Although Bid played a critical role in the caspase-1-induced apoptosis and GSDME cleavage at early time points, *Gsdmd/Bid*-DKO CL26-iCasp1 cells showed caspase-3 and caspase-9 activation, GSDME cleavage, PS exposure, plasma membrane blebbing, and LDH release when the AP20187 treatment was prolonged (Fig. 8a-c and Supplementary Fig. 12a-c). Depletion of caspase-3 from the CL26-iCasp1 cells diminished GSDME cleavage, PS exposure, LDH release, and loss of viability (Fig. 8b and Supplementary Fig. 12b-d). These observations suggested that a Bid-independent pathway exists that transduces caspase-1 activation to apoptosis in a caspase-3-dependent manner.

Regarding the specificity of the effect of CA-074Me, this compound exhibited no effect on the production of tBid and caspase-3 activation, and only marginally affected LDH release in *Gsdmd*-KO CL26-iCasp1 cells (Fig. 8f,g), indicating that CA-074Me has no or little effect on the Bid-dependent pathway and caspase-1 activity. Bafilomycin A1, like CA074Me, suppressed caspase-3 activation in AP20187-treated *Gsdmd/Bid*-DKO CL26-iCasp1 cells: this result is consistent with the idea that the Bid-independent pathway involves lysosomal proteases. In addition, bafilomycin A1, like CA074Me, did not affect tBid production in *Gsdmd*-KO CL26-iCasp1 cells, indicating that lysosomal proteases are not involved in caspase-1-mediated Bid processing. However, bafilomycin A1, unlike CA074Me, significantly reduced the activation of caspase-3 seen in *Gsdmd*-KO CL26-iCasp1 cells treated with AP20187 for 1.5 h. Therefore, we could not exclude a possibility that CA074Me and bafilomycin A1 suppressed caspase-3 activation in *Gsdmd/Bid*-DKO CL26-iCasp1 cells by mutually different mechanisms. For this reason, the results with bafilomycin A1 are not included in the present version of manuscript.

Comment #3. Please include molecular weight markers on all western blots to help readers gauge their accuracy.

Answer to comment #3:

In the original version of our manuscript, uncropped scans of Western blots with molecular weight markers are shown in the Supplementary information. According to the reviewer's suggestion, we now added molecular weight markers to all of the Western blot figures in the new manuscript.

Reviewer #2:

GSDMD is the recently identified executor of pyroptosis. It has however been known that GSDMD^{-/-} BMDMs nevertheless undergo a cell death program, which involves the activation of apoptotic caspases by caspase-1 and the induction of late necrosis (PMID: 26611636, 29281832). How exactly caspase-1 activates apoptotic caspases and executes this apoptotic program was unknown, although some reports indicated that caspase-1 might directly cleave caspase-3 (PMID: 28392147). Here the authors however show that Caspase-1 induces apoptosis (and late necrosis) in absence of GSDMD, and that this not only involves the activation of caspase-3, but all apoptotic caspases. Critical appears to be the activation of Caspase-9 and the formation of the apoptosome complex, since in absence of Caspase-9, Casp-3 is no longer processed. Their data suggest that Caspase-1 cleaves Bid, which then drives cytochrome C release and apoptosome assembly. They furthermore show that even in *Gsdmd/Bid*-DKO another pathway is activated, which causes apoptosis.

This is a solid study that investigates a timely topic, ie. the nature of GSDMD independent cell death observed after caspase-1 activation. The conclusion are for the most part justified and the paper will be an interesting manuscript for Nat Comms with some additional experiment, in particular when verifying the results in a more physiological setting. To improve the manuscript, I would suggest to address the following points:

Comment #1. The authors should provide additional data to support that caspase-1 cleaves Bid to induce mitochondrial outer membrane permeabilization and apoptosome assembly, such as showing mitochondria membrane depolarization and Apaf-1/Casp-9 oligomerization. They should also demonstrate that this involves Bax/Bak.

Answer to comment #1:

First of all, authors appreciate the reviewer for his/her positive rating of our manuscript and for valuable suggestions to improve our manuscript.

We demonstrated cytochrome c release into the cytosol as evidence for mitochondrial outer membrane permeabilization in *Gsdmd*-KO CL26-iCasp1 cells after AP20187 treatment (Fig. 6a). The cytochrome c-release was not observed in *Gsdmd/Bid*-DKO cells, indicating that it was a Bid-dependent event. In addition, we performed new experiments in which mitochondrial membrane potential was assessed by flow cytometry using TMRM. After AP20187 treatment, mitochondrial membrane potential was significantly lower in *Gsdmd*-KO cells than in *Gsdmd/Bid*-DKO cells, suggesting that Bid mediates MOMP during caspase-1-induced apoptosis (new Supplementary Fig. 8a). Unexpectedly, AP20187 treatment increased TMRM fluorescence intensity in *Gsdmd/Bid*-DKO cells. Therefore, relative TMRM fluorescence levels of *Gsdmd*-KO cells to those of *Gsdmd/Bid*-DKO cells are shown.

To examine whether the Apaf-1-Casp-9 complex is formed during caspase-1-induced apoptosis, we performed new experiments in which caspase-9 was immunoprecipitated, and the amount of Apaf-1 in the precipitants was determined by Western blotting (new Supplementary Fig. 8b). Apaf-1 was co-precipitated with caspase-9 in lysates from *Gsdmd*-KO cells treated with AP20187. On the other hand, the Apaf-1-caspase-9 complex was not formed in WT cells and *Gsdmd/Bid*-DKO cells, suggesting that caspase-1 induction of apoptosome assembly requires Bid and is suppressed in the presence of GSDMD.

To examine the involvement of Bax/Bak in caspase-1-induced apoptosis, we performed new experiments in which these genes in *Gsdmd*-KO CL26-iCasp1 cells were silenced by using siRNAs (new Supplementary Fig. 8c-f). Bax and Bak were successfully knocked down in the cells, which resulted in a reduction in caspase-3 activation after AP20187 treatment without affecting the expression and processing of Bid. These results indicate that Bax and Bak are involved in caspase-1-induced apoptosis.

These results are now described in the new manuscript as follows:

(Page 10, lines 18-21) Cytochrome c release into the cytosol, mitochondrial depolarization, and formation of an Apaf-1-caspase-9 complex were observed in *Gsdmd*-KO CL26-iCasp1 cells after AP20187 treatment (Fig. 6a and Supplementary Fig. 8a, b).

(Page 10, lines 25 – Page 11, line 10) Remarkably, cytochrome c release, mitochondrial depolarization, and Apaf-1-caspase-9 complex formation were not induced by AP20187 until 1 h after AP20187 treatment if Bid was ablated (Fig. 6a, Supplementary Fig. 8a, b, and Supplementary Fig. 9a). Moreover, the depletion of Bid severely impaired the LDH release from *Gsdmd*-KO CL26-iCasp1 cells at least up to 4 h after AP20187 treatment (Fig. 6c). Under the same conditions, caspase-3 activation and PS exposure were also severely diminished until 1.5 h after AP20187 treatment (Fig. 6d and Supplementary Fig. 9b, c). Furthermore, knockdown of Bax/Bak, pro-apoptotic partners of Bid, significantly reduced the activation of caspase-3 (Supplementary Fig. 8c-f) without affecting tBid generation. These results indicate that Bid plays a critical role in the caspase-1-induced cell death through the intrinsic apoptosis pathway in the absence of GSDMD.

Comment #2. The microscopy is poor. Often the morphological changes, that the authors refer to, are not visible in the figures. I would suggest to repeat the imaging using a better microscope, and to include galleries and close-ups of individual cells that clearly show the respective morphology.

Answer to comment #2:

We tried to improve the quality of the microscopic data. The experiments were repeated, and microscopic images have been replaced with those newly taken (Fig. 1c, e, Fig. 3e, Fig. 7c, f). Magnified images of representative cells are also shown in Fig. 1c, e, Fig. 2f, Fig. 3e, Fig. 7c, f.

Comment #3. Most of the study has been done with CL26 cells expressing chemically activatable caspase-1. The CL26 cells are a colorectal carcinoma cells, and neither used by other researchers in the field nor very physiological. This casts some doubt on the physiological significance of the findings. The results need to be verified in mouse BMDMs, which is the only universally accepted model cell line for murine inflammasome activation. Furthermore, they should show the expression levels of all inflammasome and apoptotic signaling components and compare this to levels found in BMDMs (such as GSDMD, GSDME, ASC and others).

Answer to comment #3:

We cannot completely agree with the notion that macrophages are the only accepted model for inflammasome activation, as inflammasomes have also been reported to have physiological and pathophysiological functions in dendritic cells, neutrophils, and non-myeloid cells including intestinal epithelial cells, CD4⁺ T cells and keratinocytes (PMID: 27846608, 28636595, 28410991, 24356306, 29348630). Nonetheless, as the reviewer mentioned, macrophages infected by pathogens or treated with PAMPs/DAMPs have been widely used for inflammasome and pyroptosis studies. However, the major drawback of using such experimental system is variety of intracellular signals for cell death would be activated simultaneously. This would complicate the interpretation of experimental results. In this study, we succeeded to examine simply the molecular mechanisms of caspase-1 initiated apoptosis by directly activating caspase-1 using an inducible oligomerization system.

Then we confirmed the most important findings of our study that Bid plays important role in the caspase-1-mediated apoptosis in the absence of GSDMD by using macrophages infected by bacteria or treated with an inflammasome activator. Actually, we have used BMDMs and/or Raw264.7 macrophage cells that have been widely used for inflammasome and pyroptosis studies, and demonstrated that *Salmonella* infection (NLRC4 inflammasome activator) and/or Val-boroPro (NLRP1 inflammasome activator) treatment induced caspase-1-mediated apoptosis in a manner dependent on caspase-3 and Bid in the absence of GSDMD (Fig. 1, Supplementary Fig. 2, Fig. 3d, e, Supplementary Fig. 4c, Fig. 4e, Fig. 7). In addition, we performed new experiments in which *Salmonella* growth in macrophages were examined (new Supplementary Fig. 11). The results indicate that caspase-1 deficiency as well as GSDMD/Bid double deficiency resulted in the exaggeration of intracellular bacterial growth compared to GSDMD single deficiency, suggesting that caspase-1 and Bid-dependent apoptosis plays a role in the suppression of intracellular bacterial growth in the absence of GSDMD.

On the other hand, we have to admit that we have not been able to examine the Bid-independent apoptosis using macrophages. It seems like that the involvement of the Bid-independent pathway in caspase-1-mediated apoptosis was limited in Raw264.7 cells, as *Gsdmd/Bid*-DKO Raw264.7 cells were highly protected from cell death even at 24 h after the Val-boroPro treatment (Supplementary Fig. 10d). We started to generate *Gsdmd/Bid*-DKO mice to examine the Bid-independent apoptosis using BMDMs. However, we cannot include the results in our current manuscript for timely publication.

These points are now described in the new manuscript as follows:
(Page 12, line 6-13) Then we tested the effect of caspase-1-induced apoptosis on the fate of *S.*

S. Typhimurium in macrophages, the number of intracellular bacteria was determined by gentamicin protection assay (Supplementary Fig. 11). *S. Typhimurium* persisted in Casp1^{-/-} BMMs more than in WT and Gsdmd^{-/-} BMMs (Supplementary Fig. 11a). *S. Typhimurium* could grow in *Gsdmd/Casp1*-DKO and *Gsdmd/Bid*-DKO RAW264.7 cells but not in WT and *Gsdmd*-KO RAW264.7 cells (Supplementary Fig. 11b). These results suggest that caspase-1-induced Bid-dependent apoptosis can contribute to the suppression of intracellular bacterial growth/survival in the absence of GSDMD.

(Page 13, line 5-9) In Raw264.7 cells, unlike in CL26-iCasp1 cells, the involvement of the Bid-independent pathway in caspase-1-mediated apoptosis might be limited as *Gsdmd/Bid*-DKO Raw264.7 cells were highly protected from cell death even at 24 h after the Val-boroPro treatment (Supplementary Fig. 10d). Further experiments using *Gsdmd/Bid*-DKO mice are required to evaluate the importance of the Bid-independent pathway in macrophages.

We also performed new experiments in which GSDMD, GSDME, ASC, caspase-1, and Bid protein levels were analyzed by Western blotting, and the results are shown in new Supplementary Fig. 1.

Comment #4. I am very skeptical about the proposed involvement of GSDME in causing late necrosis after activation of Caspase-1 in GSDMD^{-/-} cells. Although I agree with that GSDME can be cleaved and activated by apoptotic caspases, recent data published by N. Kayagaki suggest that GSDME is not involved in causing late necrosis (PMID: 29491424). Our own results support this notion, and it appears that BMDMs express too little GSDME, so that even if it is cleaved as shown by Lee et al. (PMID: 29491424), it will not be able to induce necrosis. Consistently, Feng Shao's group has reported, that only cell lines with high GSDME expression levels can induce GSDME-dependent necrosis (PMID: 28459430). In this regards it would be essential to know if CL26 cells express high GSDME levels. Furthermore, it is crucial for the conclusions that the authors want to take (see Figure 3f) to test BMDMs from GSDMD/GSDME dKO mice.

Answer to comment #4:

Lee et al demonstrated that GSDME is dispensable for not only flagellin-induced but also Fas ligand-induced and cytochrome c-induced secondary necrosis of immortalized BMDM cell line, suggesting that GSDME is not essential for the secondary necrosis of apoptotic macrophages. In contrast, Rogers et al. have demonstrated that depletion of GSDME reduced secondary necrosis in apoptotic BMDMs infected by VSV or treated with etoposide (PMID: 28045099). Therefore, the contribution of GSDME in the secondary necrosis of apoptotic macrophages is controversial. Wang et al (Feng Shao's group, PMID: 28459430) reported GSDME-high cells directly exhibited pyroptotic morphology, while GSDME-low cells exhibited apoptotic morphology before they became necrotic. Therefore, their results not necessarily indicate high GSDME expression is required for GSDME-mediated secondary necrosis.

We have shown that mouse BMDMs expressed a detectable level of GSDME, and processing of GSDME was observed in GSDMD-deficient but not in wild-type macrophages after *Salmonella* infection (Supplementary Fig. 5h). Furthermore, GSDME processing was less prominent in caspase-1-deficient BMDMs compared to GSDMD-deficient BMDMs under the same conditions (Supplementary Fig. 5h). Therefore, it is possible that the contribution of GSDME in the secondary necrosis of caspase-1-deficient macrophages, in which inflammasome activation would elicit caspase-8-dependent apoptosis, is limited.

As the reviewer anticipated, BMDMs expressed a lower level of GSDME protein than CL26-iCasp1 cells (new Supplementary Fig. 1). Raw264.7 cells expressed little GSDME, and

thus this cell line is not suitable to test the role of GSDME in the secondary necrosis/pyroptosis. We agree that we need to test BMDMs from *Gsdmd/Gsdme*-DKO mice to clarify whether GSDME plays important role in secondary necrosis/pyroptosis of apoptotic macrophages. However, we cannot include the results in our current manuscript for timely publication.

These points are now described in the new manuscript as follows:
(Page 9, lines 6-12) GSDME was also cleaved in GSDMD-deficient BMMs after *S. Typhimurium* infection (Supplementary Fig. 5h). However, the role of GSDME in BMM apoptosis has been controversial^{39,41}. Raw264.7 cells expressed little GSDME (Supplementary Fig. 1), and thus this cell line is not suitable to test the role of GSDME in the secondary necrosis/pyroptosis. Thus, further experiments using BMMs from *Gsdmd/Gsdme*-DKO mice are required to determine the contribution of GSDME in the secondary necrosis/pyroptosis of GSDMD-deficient macrophages.

Comment #5. The authors should analyze either combined supernatant+lysates samples, or show the supernatant samples in addition to the cell lysates. In many of their blots, such as 1d, Casp-1 p10 or Casp3 p17 is not seen in WT cells. Presumably it is made, but not visible since it is released into the supernatant upon cell lysis.

Answer to comment #5:

We performed new experiments in which proteins in culture supernatants were precipitated by the TCA method, and culture supernatants plus cell lysates (SUP+CL) were subjected to Western blotting. The results are shown in new Fig. 1d, g, and new Supplementary Fig. 3d. Caspase-1 p10 was detected in SUP+CL of WT BMMs infected with *S. Typhimurium*, suggesting that proteins were successfully precipitated. On the other hand, caspase-3 p17 was not detected in SUP+CL of WT cells (BMDMs, RAW264.7 cell, and CL26-Casp1 cells) collected after caspase-1 activation. These results further support the idea that GSDMD inhibits caspase-1-induced apoptosis.

Comment #6. GSDME cleavage blots will need to be shown in most of the figures in addition to the caspase and bid cleavage blots.

Answer to comment #6:

According to this comment, we performed new Western blotting with anti-GSDME antibody. As expected, GSDME was cleaved during caspase-1-induced apoptosis in a caspase-3-dependent manner (new Fig.3a, new Supplementary Fig. 5h, Supplementary Fig. 12b). Bid and caspase-9 were also involved in GSDME-cleavage induction (Fig. 5a, Fig. 8a), while caspase-2, 6, and 7 were dispensable for it (data not shown).

These points are now described in the new manuscript as follows:
(Page 8, line 20-21) Indeed, GSDME was cleaved in a caspase-3-dependent manner in *Gsdmd*-KO CL26-iCasp1 cells after AP20187 treatment (Fig. 3a).

(Page 9, line 6-7) GSDME was also cleaved in GSDMD-deficient BMMs after *S. Typhimurium* infection (Supplementary Fig. 5h).

(Page 10, line 9-12) By contrast, depleting caspase-9 from the *Gsdmd*-KO CL26-iCasp1 cells markedly reduced the activation of caspase-3 and GSDME, PS exposure, and LDH release induced by AP20187 treatment (Fig. 5a-d),

(Page 12, line 16-19) Although Bid played a critical role in the caspase-1-induced apoptosis and GSDME cleavage at early time points, *Gsdmd/Bid*-DKO CL26-iCasp1 cells showed caspase-3 and caspase-9 activation, GSDME cleavage, PS exposure, plasma membrane blebbing, and LDH release when the AP20187 treatment was prolonged (Fig. 8a-c and Supplementary Fig. 12a-c).

Comment #7. As for Figure 4: How do are the different caspases (Casp-2 and Casp-8) activated in the context of AP20187 treatment?

Answer to comment #7:

It appears that caspase-2 and caspase-8 were activated downstream of caspase-3, as caspase-2 p30 and caspase-8 p18 were detected in *Gsdmd*-KO cells but not *Gsdmd/Casp3*-DKO cells (new Fig. ?, Fig. 4c, d, e). Since caspase-3 has been suggested to activate other apoptotic caspases, including caspase-2 and caspase-8, directly and indirectly (PMID: 17013758, 11030145, 10578181), it is conceivable that these caspases are activated during caspase-1-induced apoptosis owing to caspase-3-mediated feedback activation. On the other hand, activation of caspase-7 as well as that of caspase-9 was partially affected by the absence of caspase-3 (new Fig. 4c, e). It has been reported that caspase-7 is cleaved by caspase-1 (PMID: 18667412). Hence, caspase-7 activation might be due to multiple mechanisms, such as processing by caspase-1 and feedback activation mediated by caspase-3.

These points are now described in the new manuscript as follows:

(Page 9, line 17 – Page 10, line 2) The cleavage of caspase-8 (generation of the p18 fragment) and caspase-2 (generation of the p30 fragment) observed in *Gsdmd*-KO CL26-iCasp1 after AP20187 treatment and in RAW264.7 cells after *S. Typhimurium* infection was almost completely abrogated in the *Gsdmd/Casp3*-DKO cell lines (Fig. 4c-e). The cleavage of caspase-7 (generation of the p20 fragment) and caspase-9 (generation of the p37 fragment) was also reduced in the *Gsdmd/Casp3*-DKO cells compared to *Gsdmd*-KO cells; however, residual activation of caspase-7 and -9 was reproducibly observed in the *Gsdmd/Casp3*-DKO cells (Fig. 4c, e). These results indicated that caspase-2 and -8 are activated downstream of caspase-3, whereas caspase-7 and -9 can be activated both independently of (upstream of or in parallel with) and downstream of caspase-3 in *Gsdmd*-KO cells under these experimental conditions.

Minor comments:

1. Some of the blots need better loading controls, such as 1e and others.

Answer to minor comment #1:

We performed new Western blot experiments corresponding to the original Fig. 1d, e using SUP+CL samples as described in the answer to the comment #5. Accordingly, original Fig. 1d, e are replaced with new combined Fig. 1d. For Fig. 2c, Fig. 3a, Fig. 4c, Fig. 6b, e, we guessed that the unclear GAPDH bands were due to excessive amount of GAPDH protein in each lane. Therefore, we have repeated Western blotting, in which reduced volume of the same samples were loaded, and blots with unclear GAPDH bands have been replaced with new ones.

2. Verify the specificity of the Bid inhibitor.

Answer to minor comment #2:

To verify the specificity of the Bid inhibitor BI-6C9, we used *Gsdmd*-KO RAW264.7 cells and *Gsdmd/Bid*-DKO RAW264.7 cells. Staurosporine induced caspase-3 activation to a similar extent in both cell lines, suggesting that Bid is dispensable for the response. Unexpectedly, BI-

6C9 reduced staurosporine-induced activation of caspase-3 in both cell lines. Therefore, the specificity of BI-6C9 is unclear. Therefore, we removed experimental results involving BI-6C9 from our manuscript.

Then, we performed new experiments in which Bid expression in *Gsdmd*-KO BMDMs was knocked down using siRNAs (new Fig. 7b, c). This resulted in the reduction of *S. Typhimurium*-induced caspase-3 activation and apoptosis in this cell line. These results suggest that Bid is involved in caspase-1-induced apoptosis in *Gsdmd*-KO BMDMs.

These results are now described in the new manuscript as follows:
(Page 11, line 18-19) Knockdown of Bid suppressed the *S. Typhimurium*-induced caspase-3 activation (Fig. 7b), and diminished the apoptotic morphological changes in *Gsdmd*^{-/-} BMDMs (Fig. 7c).

3. Taabazuing et al. have shown that Caspase-1 can directly cleave and activate Caspase-3, this should be more clearly discussed in the text.

Answer to minor comment #3:

Taabazuing et al. (PMID: 28392147) showed that recombinant caspase-1 cleaved pro-caspase-3 and -7 in THP1 cell lysate (a cell free system), but it was not tested whether the cleaved caspases were functional and whether the direct cleavage really happened in cells were not examined. Therefore, Taabazuing et al. themselves described in their paper that “We therefore speculate that active caspase-1 directly cleaves caspase-3/-7 to trigger apoptosis, although direct in vivo cleavage remains to be definitively demonstrated.

These points are now described in the new manuscript as follows:
(Page 14, line 21 – Page 15, line 1) Taabazuing et al. demonstrated that Val-boroPro, a dipeptidyl peptidase (DPP) inhibitor, which induce a caspase-1-dependent cell death, induced apoptosis accompanied by caspase-3 and caspase-7 activation in GSDMD-deficient RAW264.7 and THP-1 cells³². They also demonstrated that recombinant caspase-1 cleaved caspase-3 and caspase-7 in a cell free system. However, whether the cleaved caspases were functional and whether the direct cleavage really happened in cells were not examined.

Reviewer #3:

Summary/Major Criticism: In this manuscript, the authors show that in the presence of inflammasome stimuli and the absence of gasdermin D, caspase-1 leads to the activation of classical markers of apoptosis. Most notably, caspase -1 leads to caspase-3 cleavage via the activation of Bid/caspase-9. This leads to apoptosis as well as secondary necrosis mediated by GSDME. The authors’ data also suggest caspase-1 activation in the absence of GSDMD leads to a Bid-independent apoptosis pathway, which remains to be elucidated. Although this manuscript takes a detailed mechanistic approach to dissect caspase-1- induced apoptosis, the authors mainly use a highly engineered cell line and the relevancy of this pathway is not demonstrated. Given that other recent publications (Lee et al., 2018, Scientific Reports, 8:3788 and Schneider et al., 2017, Cell Reports 21, 3846-3859) have described caspase-1-mediated apoptosis without identifying a physiologically relevant cell type/disease context, it is unclear how this study expands on these previous results. The authors even identify cells where this cell death pathway may occur but do not go on to test whether caspase-1-mediated apoptosis actually occurs in these cells. Thus, a major weakness of this manuscript is the lack of evidence of physiological significance. If the authors were to demonstrate that caspase-1- mediated

apoptosis occurs via Bid/caspase-9/caspase-3 in a “wild-type” cell type, this would demonstrate relevancy and significance of the cell death pathway they describe.

Answer to the major criticism.

First of all, authors appreciate for valuable reviewer’s suggestions to improve our manuscript.

To address this criticism, we obtained SH-SY5Y human neuroblastoma cells, which is recently reported to exhibit caspase-1-inhibitor sensitive apoptosis in response to 1-methyl-4-phenylpyridinium ion treatment (PMID: 27339879). However, unfortunately, the obtained SH-SY5Y cells did not express detectable levels of caspase-1. Because genetic engineering experiments using SH-SY5Y cells are not familiar to us, we then employed L929 cells as wild-type cells that do not express GSDMD to generate Fv3-Casp1 transfectants (L929-iCasp1 cells, Supplementary Fig. 1 and Fig. 9). AP20187 treatment induced tBid generation, caspase-9 activation, GSDME cleavage, PS exposure before membrane permeabilization, membrane blebbing, indicating that AP20187 induced apoptosis in L929-iCasp1 cells. Furthermore, Transduction of GSDMD into L929-iCasp1 cells converted the mode of caspase-1-initiated cell death from apoptosis to pyroptosis. (Fig. 9c-d). These results indicate that caspase-1 activation (although in an artificial way) can initiate apoptosis in “wild-type” cells when they do not express GSDMD.

These results are now described in the new manuscript as follows:

(Page 13, line 24 – page 14, line 4) To test this possibility, we have employed L929 cell that do not express GSDMD (Supplementary Fig. 1). L929 cells transduced with Fv3-mCasp1 (L929-iCasp1 cells) underwent apoptosis after AP20187 treatment (Fig. 9a-c). Moreover, ectopic expression of GSDMD changed the form of cell death in the L929-iCasp1 cells into pyroptosis (Fig. 9c-e), suggesting that caspase-1 induces apoptosis but not pyroptosis in L929 cells, most likely because the cells do not express GSDMD. Accordingly, the type of cell death induced after caspase-1 activation in non-myeloid cells should be carefully examined, as it could be pyroptosis or apoptosis, depending on the expression of GSDMD.

Major comments:

Comment #1. For the in vitro *Salmonella* infection assays, can the authors please explain why they used streptomycin/penicillin to kill *Salmonella* after infection instead of the standard gentamicin? Additionally, the pictures in Fig. 1c,f, 3e, and 7c,f appear to show high concentrations of extracellular *Salmonella*. Standard in vitro *Salmonella* infection assays involve washing the cells to remove extracellular bacteria before adding on fresh media with antibiotics to maintain only intracellular infection. Can the authors please clarify if they performed a wash step? High amounts of extracellular bacteria can contribute to LDH signal, making removal of excess extracellular bacteria critical for LDH assays. If the authors did not follow standard gentamicin protection assays, the authors should repeat the *Salmonella* infection experiments using this standard method to ensure their current results correlate with the field standard for this type of assay.

Answer to comment #1:

In the original version of manuscript, *Salmonella* was not washed out before addition of penicillin/streptomycin. According to this reviewer’s comment, we performed new experiments in which cells were infected with *Salmonella* in antibiotic-free medium, washed to remove extracellular bacteria, and then incubated in the presence of gentamicin (new Fig. 1c, e, Fig. 3e, Fig. 7c, f). The results are consistent with those in the original manuscript. In new Fig. 1b, d, bacteria were not washed out, since cell death was induced quite rapidly after *Salmonella* infection and culture supernatants were collected as samples. Gentamicin was added to the cultures 1 h after *Salmonella* infection in Fig. 1b, d. Instead of washing,

macrophage-free wells were made to monitor LDH release from *Salmonella* (Fig. 1b). LDH activity in macrophage-free wells was much lower than that in wells of *Salmonella*-infected macrophages, indicating that levels of LDH from *Salmonella* were ignorable. Western blotting experiments, except for Fig. 1d, were not repeated, because most of the samples were collected before the time of washing and addition of antibiotics.

Accordingly, the description in the Methods section was revised as follows:
(Page 20, line 18-22) BMMs and RAW264 cells cultured in antibiotic-free RPMI1640 medium at a density of 5×10^4 /well in 96-well plates were infected with *S. Typhimurium* at an MOI of 5 and 40, respectively. The plates were centrifuged at 2500 rpm for 5 min and then transferred to a CO₂ incubator. After 1 h (BMMs) or 2 h (RAW264 cells), the cells were washed three times with antibiotic-free medium, and gentamicin (50 µg/ml) was added to the wells.

Comment #2. For the in vitro *Salmonella* assays, did the authors quantify levels of intracellular bacteria? Inflammasome activation and pyroptosis are important innate defense mechanisms that restrict intracellular pathogens. Providing CFU counts/bacterial replication assay data at different time points for the intracellular *Salmonella* infections may help the authors demonstrate when the different pathways they have identified play a role in a physiologically relevant scenario.

Answer to comment #2:

We appreciate this suggestion that helps to increase the significance of our findings. We performed new experiments in which intracellular bacteria levels were quantified by gentamicin protection assay (new Supplementary Fig. 11b). In these experiments, BMMs and RAW264.7 cells were infected with *Salmonella*, and 1 or 2 h later, cells were washed and gentamicin was added. Under these conditions, the bacterium persisted in caspase-1-deficient BMMs more than in WT and GSDMD-deficient BMMs. *Salmonella* could grow in *Gsdmd/Casp1*-DKO and *Gsdmd/Bid*-DKO RAW264.7 cells but not in WT and *Gsdmd*-KO RAW264.7 cells. These results suggest that caspase-1-induced apoptosis contributes to the suppression of intracellular bacterial growth in the absence of GSDMD.

These results are now described in the new manuscript as follows:
(Page 12, lines 6-13) Then we tested the effect of caspase-1-induced apoptosis on the fate of *S. Typhimurium* in macrophages, the number of intracellular bacteria was determined by gentamicin protection assay (Supplementary Fig. 11). *S. Typhimurium* persisted in *Casp1*^{-/-} BMMs more than in WT and *Gsdmd*^{-/-} BMMs (Supplementary Fig. 11a). *S. Typhimurium* could grow in *Gsdmd/Casp1*-DKO and *Gsdmd/Bid*-DKO RAW264.7 cells but not in WT and *Gsdmd*-KO RAW264.7 cells (Supplementary Fig. 11b). These results suggest that caspase-1-induced Bid-dependent apoptosis can contribute to the suppression of intracellular bacterial growth/survival in the absence of GSDMD.

Comment #3. For the caspase-9 experiments, did the authors perform complementation of the caspase-9 knockout cells to show restoration of the cell death phenotype? If so, these data should be included in the supplemental figures.

Answer to comment #3:

According to this comment, *Casp9*-KO cells transduced with the *Casp9* cDNA were generated to confirm the role of caspase-9 in caspase-1-induced apoptosis (new Fig. 5e and Supplementary Fig. 7). Lentiviral expression of this caspase in *Casp9*-KO CL26-iCasp1 cells restored phosphatidylserine exposure, caspase-3 activation, and LDH release.

These results are now described in the new manuscript as follows:
(Page 10, lines 9-14) By contrast, depleting caspase-9 from the *Gsdmd*-KO CL26-iCasp1 cells markedly reduced the activation of caspase-3 and GSDME, PS exposure, and LDH release induced by AP20187 treatment (Fig. 5a-d), which were restored by complementation with the Casp9 cDNA (Fig. 5e and Supplementary Fig. 7a, b), indicating that caspase-9 mediates the apoptosis initiated by caspase-1.

Comment #4. The authors frequently reference LDH release as evidence of apoptosis. However, LDH is released during pyroptotic and necroptotic cell death, not apoptotic cell death. How do the authors explain using LDH as a measure of apoptotic cell death? When the authors knock out *Gsdme*, they see reduced levels of LDH release. Since GSDME has been shown to mediate secondary necrosis, these data suggest the authors are observing secondary necrosis as measured by LDH release as well as apoptosis as measured by phosphatidylserine exposure simultaneously.

Answer to comment #4:

Kayagaki et al previously demonstrated that ATP-induced NLRP3 inflammasome activation induces LDH release in Pam3CSK-primed *Gsdmd*-KO but not Casp1-KO BMDMs (PMID: 26375259). They suggested that there is a Caspase-1-dependent but GSDMD-independent pyroptosis pathway. We used LDH release assay to demonstrate that the observed caspase-1-dependent apoptosis could eventually cause LDH release, and thus it can explain the phenomenon that was observed by Kayagaki et al.

We agree that LDH release alone is not appropriate as a measure of apoptotic cell death. However, LDH release can occur as a consequence of secondary necrosis of apoptotic cells *in vitro*. Therefore, the LDH release can be used as a measure of *in vitro* apoptosis when the cell death mode is shown to be apoptosis. We used morphological features (membrane blebbing and formation of apoptotic bodies), PS exposure (annexin V-staining) before membrane permeabilization (propidium iodide-staining), and caspase-3-dependency as evidence for apoptosis throughout our paper. In addition, LDH release assay was useful to measure both pyroptosis and apoptosis by a single assay.

Comment #5. The authors state that previous studies showed caspase-3 can cleave GSDME. Given that the authors showed GSDME mediates LDH release and secondary necrosis in their system, did the authors look for GSDME cleavage in the presence/absence of caspase-3, caspase-9, and Bid? These data would strengthen their argument and further elucidate this pathway.

Answer to comment #5:

To address this comment, we performed new Western blot experiments with anti-GSDME antibody. As expected, GSDME was cleaved during caspase-1-induced apoptosis in a caspase-3-dependent manner (Fig.3a, Supplementary Fig. 5b, Supplementary Fig. 12b). Caspase-9 was also involved in inducing GSDME cleavage (Fig. 5a), while caspase-2, 6, and 7 were dispensable for it (data not shown). Bid was required for GSDME cleavage (Fig. 8a, at 45 and 90 min). However, GSDME was cleaved by a Bid-independent mechanism at 240 min as described in the answer to comment #6.

These results are now described in the new manuscript as follows:
(Page 8, line 20-21) Indeed, GSDME was cleaved in a caspase-3-dependent manner in *Gsdmd*-KO CL26-iCasp1 cells after AP20187 treatment (Fig. 3a).

(Page 10, line 9-12) By contrast, depleting caspase-9 from the *Gsdmd*-KO CL26-iCasp1 cells markedly reduced the activation of caspase-3 and GSDME, PS exposure, and LDH release induced by AP20187 treatment (Fig. 5a-d),

(Page 12, line 16-19) Although Bid played a critical role in the caspase-1-induced apoptosis and GSDME cleavage at early time points, *Gsdmd/Bid*-DKO CL26-iCasp1 cells showed caspase-3 and caspase-9 activation, GSDME cleavage, PS exposure, plasma membrane blebbing, and LDH release when the AP20187 treatment was prolonged (Fig. 8a-c and Supplementary Fig. 12a-c).

Comment #6. Based on the data in Figure 8, the authors suggest there is a Bid independent apoptosis pathway that occurs at later time points (i.e., 8 hr instead of 4 hr). However, this conclusion is mostly based on LDH data, which is indicative of pyroptosis/necroptosis. Given that the authors looked at the contribution of GSDME in their system in a previous figure, can the authors please explain why they did not examine the role of GSDME in this Bid - independent pathway?

Answer to comment #6:

We have shown that the Bid-independent cell death was accompanied with caspase-3 activation and phosphatidylserine exposure (Fig. 8a, b). We also performed new experiments, and demonstrated caspase-3 dependency and apoptotic morphology of caspase-1-initiated cell death of *Gsdmd/Bid*-DKO CL26iCasp1 cells at 4 h after AP20187 treatment (new Fig. 8b, Supplementary Fig. 12a), further supporting the notion that the Bid-independent cell death was apoptosis. In addition, as described above, GSDME was cleaved by a Bid-independent but caspase-3-dependnet mechanism at 4 h after AP20187 treatment (new Fig. 8a, Supplementary Fig. 12b). Consistently, LDH release from *Gsdmd/Bid*-DKO cells at 24 h after AP20187 treatment was diminished by further depletion of Caspase-3 (new supplementary Fig12c). We also confirmed that the Bid-independent cell death was caspase-3 dependent by using an alternative cell death assay, WST-1 assay (new Supplementary Fig. 12d).

These results are now described in the new manuscript as follows:
(Page 12, line 17-23) ..., *Gsdmd/Bid*-DKO CL26-iCasp1 cells showed caspase-3 and caspase-9 activation, GSDME cleavage, PS exposure, plasma membrane blebbing, and LDH release when the AP20187 treatment was prolonged (Fig. 8a-c and Supplementary Fig. 12a-c). Depletion of caspase-3 from the CL26-iCasp1 cells diminished GSDME cleavage, PS exposure, LDH release, and loss of viability (Fig. 8b and Supplementary Fig. 12b-d). These observations suggested that a Bid-independent pathway exists that transduces caspase-1 activation to apoptosis in a caspase-3-dependent manner.

Minor comments:

1. The authors should recheck the manuscript for spelling and grammar errors (e.g., Salmonella spelled Salomonella or caspase spelled capase or caspas).
2. Page 13, line 23, “Ding et al. reported...” This citation reference is wrong. The paper the authors are referencing is He et al. and is listed as reference number 20 in their references list.
3. In the Results section titled “Caspase-1-induced apoptosis involves caspase -9,” lines 21 and 24, the authors incorrectly refer to Fig. 4d, e and then Fig. 4f, g when they should refer to Fib. 4e, f and Fig. 4g, h, respectively.

4. In the legend for Fig. 5a, the authors state that the Western blot probes for Bid. However, the provided image does not indicate Bid was examined. The authors should correct the legend or blot image accordingly.

Answer to minor comments:

The authors apologize for our careless mistakes and appreciate your perusal of our manuscript.

1. We recheck the manuscript for spelling and grammar errors.

2. The corresponding sentence in the new manuscript was corrected as follows:

(Page 15, line 12-13) He et al²⁰ reported similar observations using ASC-reconstituted RAW264.7 cells.

3. The corresponding sentences in the new manuscript was corrected as follows (Figure numbers are changed by addition of new figures):

(Page 10, line 3-5) We next set out to identify caspases involved in the caspase-1-induced apoptosis. Depleting caspase-2, -6, or -7 from the *Gsdmd*-KO CL26-iCasp1 cells did not affect the apoptosis induced by AP20187 (Fig. 4f, g and Supplementary Fig. 6a).

(Page 10, line 7-9) The *Gsdmd/Rip3k*-DKO and *Gsdmd/Rip3k/Casp8*-triple KO cell lines underwent apoptosis normally after AP20187 treatment (Fig. 4h, i and Supplementary Fig. 6b), indicating that the caspase-1-induced apoptosis can proceed independently of these caspases.

4. Bid was removed from the legend for Fig. 5a as follows (Western blot for GSDME was added according to comment #5):

(Page 39, line 8-10) *Gsdmd*-KO and *Gsdmd/Casp9*-DKO CL26-iCasp1 cells were treated with 50 nM AP20187 for the indicated times. Western blot detection of cleaved caspases and GSDME (a).

Reviewers' comments:

Reviewer #1 (Remarks to the Author):

The authors have very nicely addressed all my queries and have substantially improved the overall quality. This manuscript will be of great interest to both the cell death and inflammasome fields.

Reviewer #2 (Remarks to the Author):

The authors have addressed my request and convincingly show that Caspase-1 can induce an "kind of" apoptotic cell death in GSDMD^{-/-} cells.

What surprises me about this is the speed by which this type of death becomes lytic, i.e. LDH release can be measured. It suggest that intrinsically this caspase-1-dependnet apoptosis is different or faster than regular apoptosis.

I think that the paper would profit from a closer examination of different features observed during 'regular' apoptosis (using FasL or other inducers) and this type of caspase-1-dependnet apoptosis. This could be done in form of a time-course analysis and in a summary table that examines different markers of apoptotic cells (i.e. morphological changes as well as the processing of certain caspases and their substrates).

Reviewer #3 (Remarks to the Author):

My major criticism of this paper was not addressed and still holds as a major impediment to publication. Briefly, there is no evidence that Casp1 initiates a form of cell death that is independent of Gsdmd in any physiologically relevant context. The authors should demonstrate that Casp1-induced apoptosis occurs in a cell type that has not been genetically altered. For example, can the authors provide evidence that spinal cord cells (or another cell type that expresses Casp1 but not Gsdmd naturally) undergo Casp1-induced apoptosis. The authors engineer L929 cells with the active Casp1 construct as a rebuttal to this criticism. However, it is not clear that L929 undergo any kind of Casp1-induced cell death naturally. Do these cells even express Casp1?

I do not recommend continued publication of research projects that characterize Casp1-induced apoptosis in the absence of Gsdmd expression without a biologically relevant context.

Point-by-point reply to the reviewers' comments

Comment from Reviewer #1:

The authors have very nicely addressed all my queries and have substantially improved the overall quality. This manuscript will be of great interest to both the cell death and inflammasome fields.

Answer to the comment from Reviewer #1

We are most grateful for your perusal of our manuscript and your positive comment for our manuscript.

Comment from Reviewer #2:

The authors have addressed my request and convincingly show that Caspase-1 can induce an "kind of" apoptotic cell death in GSDMD^{-/-} cells.

What surprises me about this is the speed by which this type of death becomes lytic, i.e. LDH release can be measured. It suggest that intrinsically this caspase-1-dependnet apoptosis is different or faster than regular apoptosis.

I think that the paper would profit from a closer examination of different features observed during 'regular' apoptosis (using FasL or other inducers) and this type of caspase-1-dependnet apoptosis. This could be done in form of a time-course analysis and in a summary table that examines different markers of apoptotic cells (i.e. morphological changes as well as the processing of certain caspases and their substrates).

Answer to the comment from Reviewer #2

To compare the speed at which caspase-1-initiated and caspase-8 initiated (regular) apoptosis proceed to the lytic (secondary necrosis) stage, we measured caspase-3/7 activity in and LDH release from cells undergoing these types of apoptosis. First, *Gsdmd*-KO bone marrow-derived macrophages (BMMs) were infected with *Salmonella* to induce caspase-1-induced apoptosis, whereas the same cells were treated with recombinant (r)FasL plus cycloheximide (CHX) to induce caspase-8-initiated apoptosis. In both conditions, the intervals of time from caspase-3/7 activation to LDH release were approximately 2 h. Furthermore, the intervals from caspase-3/7 activation to LDH release after AP20187 treatment were similar between CL26-iCasp1 cells and CL26-iCasp8 cells. Judging from the results, we do not think that caspase-1-induced apoptosis becomes lytic faster than regular apoptosis. In addition, *Salmonella* infection and rFasL+CHX treatment induced similar morphological changes and processing of caspase-7, 8, 9, Bid, and GSDME in *Gsdmd*-KO BMMs. Consistent results were obtained with *Gsdmd*-KO CL26-iCasp1 cells and CL26-iCasp8 cells treated with AP20187. Thus, we could not find any significant difference between caspase-1-induced apoptosis and regular apoptosis, except for the timing of the onset of caspase-3/7 activation. For these reasons, we did not add a summary table that Reviewer #2 requested to this manuscript. The results of the time-course analyses are now described from page 8, line 190 to page 9, line 199 of the merged manuscript PDF and the data are shown in Supplementary Fig. 4d-g of the current version of our manuscript.

In the previous comments, Reviewer #2 suggested to use *Gsdmd*/*Bid*-double knockout (DKO) BMMs for the purpose of testing the involvement of Bid in caspase-1-induced apoptosis in more physiological settings. However, the generation of the mouse strain had not

been completed at that time, and we had therefore been unable to address this point. Recently, we have generated the *Gsdmd/Bid*-DKO mouse strain. *Gsdmd*-KO BMMs and *Gsdmd/Bid*-DKO BMMs were infected with *Salmonella* to induce caspase-1-induced apoptosis. Caspase-3 activation and LDH release induced by *Salmonella* were significantly lower in *Gsdmd/Bid*-DKO BMMs than in *Gsdmd*-KO BMMs, further supporting that Bid contributes to caspase-1-induced apoptosis. These results are now described in page 12, line 276-277 of the merged manuscript PDF, and data are shown in Fig. 7d, e of the current version of our manuscript.

Comment from Reviewer #3:

My major criticism of this paper was not addressed and still holds as a major impediment to publication. Briefly, there is no evidence that Casp1 initiates a form of cell death that is independent of Gsdmd in any physiologically relevant context. The authors should demonstrate that Casp1-induced apoptosis occurs in a cell type that has not been genetically altered. For example, can the authors provide evidence that spinal cord cells (or another cell type that expresses Casp1 but not Gsdmd naturally) undergo Casp1-induced apoptosis. The authors engineer L929 cells with the active Casp1 construct as a rebuttal to this criticism. However, it is not clear that L929 undergo any kind of Casp1-induced cell death naturally. Do these cells even express Casp1?

I do not recommend continued publication of research projects that characterize Casp1-induced apoptosis in the absence of Gsdmd expression without a biologically relevant context.

Answer to the comment from Reviewer #3

A previous report has shown that oxygen/glucose deprivation (OGD) induced apoptosis in primary cortical neurons in a caspase-1-dependent manner (PNAS 2003 100:16012). Accordingly, we speculated and mentioned in our original manuscript that caspase-1-induced apoptosis may occur in neuronal cells under pathological conditions. In the present manuscript, we have tested this assumption by reproducing the results in the previous report using primary cortical neurons obtained from wild-type (WT) and *Casp1*-KO mice. Furthermore, we investigated the expression of GSDMD in WT cortical neurons and involvement of Bid using primary cortical neurons from *Bid*-KO mice. GSDMD was not detected in WT cortical neuronal cells, and the cells underwent apoptosis accompanied by caspase-3 activation after OGD. The OGD-induced apoptosis was significantly reduced in the absence of caspase-1 or Bid, suggesting that caspase-1 and Bid-dependent apoptosis can occur in neuronal cells under pathological conditions. Furthermore, GSDMD expression was significantly lower in bone marrow-derived mast cells than in BMMs. Mast cells underwent apoptosis after nigericin treatment, which activates the NLRP3 inflammasome, in a caspase-1-dependent manner. Nigericin treatment induced caspase-3 activation in mast cells from WT mice, but not those from caspase-1-deficient mice or Bid-deficient mice. These results suggest that caspase-1-induced apoptosis can also occur in mast cells. These results are now described in the abstract and in the results section from page 14, line 340 to page 15, line 358 of the merged manuscript PDF, and data are shown in Fig. 10 and Supplementary Fig. 13g,f.

REVIEWERS' COMMENTS:

Reviewer #1 (Remarks to the Author):

The authors have satisfied all the reviewer concerns, including experimental data to substantiate the idea that caspase-1 and Bid-mediated cell death may occur in cell types (such as cortical neurons and mast cells) where GSDMD levels are absent or low. If data on cell death (LDH release/PI staining) were available for GSDMD deficient neurons/mast cells (in the context of Fig. 10) then this would be worth including for the sake of completeness. However, I do not view this as essential for publication, as the current story already elegantly delineates a novel caspase-1 killing mechanisms that, in itself, is significant enough to warrant publication.

Point-by-point reply to the reviewer's comments

Reviewer #1

The authors have satisfied all the reviewer concerns, including experimental data to substantiate the idea that caspase-1 and Bid-mediated cell death may occur in cell types (such as cortical neurons and mast cells) where GSDMD levels are absent or low. If data on cell death (LDH release/PI staining) were available for GSDMD deficient neurons/mast cells (in the context of Fig. 10) then this would be worth including for the sake of completeness. However, I do not view this as essential for publication, as the current story already elegantly delineates a novel caspase-1 killing mechanisms that, in itself, is significant enough to warrant publication.

Answer to the comment from Reviewer #1

First of all, we are most grateful for your perusal of our manuscript and for your positive comment to our manuscript.

Unfortunately, it is difficult to perform new experiments using cortical neurons from day 15 mouse embryos and 4-week cultured bone marrow-derived mast cells, as we are asked to revise our manuscript within 2 weeks. Because the reviewer kindly stated that these experiments are not essential for publication, we would like to skip these experiments this time. In addition, nuclear pyknosis and fragmentation in active caspase-3-positive cortical neurons after OGD treatment (supplementary Fig. 13g) strongly suggest that caspase-3-positive cortical neurons were actually killed. PI staining for mast cells from WT or Casp1-KO mice were performed in the flow cytometry experiments shown in Fig. 10c. The bar graph/dot plot figure depicting the proportion of annexin V+ PI+ mast cells before and after nigericin treatment are now provided in Fig. 10c.